# Phenotypic profiling with a living biobank of primary rhabdomyosarcoma unravels disease heterogeneity and AKT sensitivity

Gabriele Manzella[1], Leonie D. Schreck[1], Willemijn B. Breunis[1,2], Jan Molenaar[2], Hans Merks[2], Frederic G. Barr[3], Wenyue Sun[3], Michaela Römmele[1], Luduo Zhang[1], Joelle Tchinda[1], Quy A. Ngo[1], Peter Bode [4], Olivier Delattre [5], Didier Surdez [5], Bharat Rekhi [6], Felix K. Niggli[1], Beat W. Schäfer [1,7 ✉] & Marco Wachtel[1,7]

Cancer therapy is currently shifting from broadly used cytotoxic drugs to patient-specific precision therapies. Druggable driver oncogenes, identified by molecular analyses, are present in only a subset of patients. Functional profiling of primary tumor cells could circumvent these limitations, but suitable platforms are unavailable for most cancer entities. Here, we describe an in vitro drug profiling platform for rhabdomyosarcoma (RMS), using a living biobank composed of twenty RMS patient-derived xenografts (PDX) for high-throughput drug testing. Optimized in vitro conditions preserve phenotypic and molecular characteristics of primary PDX cells and are compatible with propagation of cells directly isolated from patient tumors. Besides a heterogeneous spectrum of responses of largely patient-specific vulnerabilities, profiling with a large drug library reveals a strong sensitivity towards AKT inhibitors in a subgroup of RMS. Overall, our study highlights the feasibility of in vitro drug profiling of primary RMS for patient-specific treatment selection in a co-clinical setting.

[1] University Children's Hospital, Department of Oncology and Children's Research Center, Steinwiesstrasse 75, CH-8032 Zurich, Switzerland. [2] Princess Máxima Center for Pediatric Oncology, Uppsalalaan 8, 3584 CT Utrecht, The Netherlands. [3] Laboratory of Pathology, Center for Cancer Research, National Cancer Institute, Bethesda, MD 20892, USA. [4] University Hospital Zurich, Institute of Surgical Pathology, Schmelzbergstrasse 12, CH-8091 Zurich, Switzerland. [5] France INSERM U830, Équipe Labellisé LNCC, PSL Université, SIREDO Oncology Centre, Institut Curie, Paris, France. [6] Tata Memorial Hospital, Department of Pathology, Dr E.B. road, Parel, Mumbai 400012, India. [7] These authors contributed equally: Beat W. Schäfer, Marco Wachtel. ✉email: beat.schaefer@kispi.uzh.ch

While cancer treatment improved dramatically during the past decades and nowadays allows cures for many previously fatal cases, in a significant number of patients therapy still fails. An important reason for this variability in therapy success is inter-tumoral heterogeneity, which is a characteristic of most if not all tumor categories. Hence, it is generally believed that future therapeutic approaches should be more patient-tailored and take into account the specific molecular and cellular characteristics of the individual tumors. Along these lines, genetic analysis is now a clinical routine for different types of cancers and allows identification of patient-specific driver oncogenes that serve as relevant therapeutic targets. Unfortunately, genome-based drug selection is not feasible for tumors which are driven by mutated but undruggable targets and have an otherwise low mutational burden, a characteristic of many pediatric sarcomas and leukemias.

An example for this class of tumors is rhabdomyosarcoma (RMS), which is the most common soft-tissue sarcoma in children and is composed of several histological subtypes, all having cells with characteristics of a propensity for skeletal-muscle differentiation. The two main histological RMS subtypes, alveolar (ARMS) and embryonal (ERMS), are characterized by distinct genetic alterations. Most ARMS cases are associated with specific translocations, generating PAX3-FOXO1 or PAX7-FOXO1 fusion transcription factors that act as the main drivers of tumorigenesis. ERMS cases, in contrast, do not contain these translocations and have a more heterogeneous genetic landscape. One important subgroup is associated with mutational activation of the RAS-pathway. Based on these molecular characteristics, RMS is subclassified into fusion-positive (FP-RMS) and fusion-negative (FN-RMS) RMS. In both FP-RMS and FN-RMS, the number of driver oncogenes that hold the potential for druggability is very low[1–3]. Despite the great progress that has been made in RMS treatment by optimizing conventional therapies, up to 30 percent of patients still have dismal outcome and no targeted therapy has entered into clinical practice so far[4]. This situation highlights the unmet medical need to address alternative routes that aid cancer-drug treatment decisions.

An alternative option for identification of drugs effective against mutationally quiet tumors is direct and unbiased testing of compounds on freshly isolated, patient-derived cancer cells. In general, patient cells can be propagated in mice as patient-derived xenografts (PDX) or grown in vitro under suitable culture conditions in 2D or 3D. While PDX models constitute an important tool for expansion of patient-derived biopsies and, at least during early passages, closely resemble the original tumor specimen at the morphological and molecular level, they might not be affordable in terms of cost and space for many laboratories and are unsuitable for large high throughput drug testing[5,6]. For large phenotypic screenings with drug libraries, in vitro cultures of cancer cells are more suitable. In case of leukemia as well as some carcinoma entities such screening approaches have proven to produce data of clinical utility[7–11]. Importantly however, optimized protocols for primary cancer cell cultures have been established for only a small number of entities and not yet for sarcoma.

Here, we set out to develop a functional predictive preclinical toolkit which combines the use of PDX and primary cultures for RMS. In this respect, we present the systematic development of optimized culture conditions for PDX-derived primary RMS cells. Importantly, we detect subtype specific demands on culture parameters such as presence or absence of bFGF and find that traditional culture conditions with presence of serum have a toxic effect on primary RMS cells, leading to the outgrowth of resistant clones. Comprehensive molecular characterization demonstrates that cultured cells are stable both at the genomic and genetic level

under these conditions. Additionally, by performing drug screens we show the feasibility of our platform to pinpoint patient-specific pharmacological vulnerabilities with a high-fidelity prediction of the in vivo response and describe an AKT-inhibitor sensitivity of specific samples. Finally, we show that established culture conditions allow propagation and drug-profiling of cells directly isolated from patient tumors, considerably accelerating the procedure. Our platform therefore holds the promise to be a valid tool for prioritizing actionable drug targets for RMS in a clinical setting.

## Results

**Culture conditions for PDX-derived primary RMS cells.** To set up a cell culture system that closely preserves the phenotypic and molecular characteristics of cells in the parental tumor, we followed the scheme depicted in Fig. 1a. We used a collection of 20 RMS PDX tumors (12 FN-RMS and 8 FP-RMS), either established on our own with a success rate comparable to previous reports[12] or in one of the three collaborating institutions (Supplementary Fig. 1A, B)[12]. Then, we first aimed to determine the optimal culture conditions supporting growth of the PDX-derived primary cells (PPCs) in vitro. At least two independent pieces from each PDX model were re-transplanted s.c. into NSG mice and grown up to 1 cm$^3$, followed by tumor isolation, dissociation into a single cell suspension and culture in 96-well plate format. We compared 18 different culture conditions by combining three media (DME and F10 medium both supplemented with 10% heat inactivated fetal bovine serum [FBS] and Neurobasal [NB] medium supplemented with serum replacement B-27) with three types of adhesion substrates (no coating, Matrigel and Gelatin), each in the presence or absence of growth factors (GF) (EGF plus bFGF). Cell viability/proliferation was determined by WST-1 assay (normally within 1-3 weeks), and cell morphology, differentiation status and contamination with mouse stromal cells was assessed by microscopy (Fig. 1b). Differentiation was monitored by Myosin Heavy Chain (MHC) expression (Supplementary Fig. 1C), while mouse cells were identified by the punctate DAPI-staining pattern of their nuclei[13] (Supplementary Fig. 1C). The analysis revealed that F-10 medium was not effective in supporting PPC growth in most cases and therefore was not studied further (Fig. 1b and Supplementary Fig. 1D). In contrast, NB conditions supported growth of cells from most PDX tumors, especially in combination with an adhesion substrate, and also exhibited the lowest percentage of mouse contamination across the entire set of samples tested (Fig. 1b and Supplementary Fig. 1D). DMEM conditions typically yielded high cell viability scores independently from the coating matrix or presence of GFs (Fig. 1b and Supplementary Fig. 1D). Importantly however, morphological analysis revealed that cells in DMEM often had an increased size and were rich in stress-fibers, suggesting that this conditions does not support continuous proliferation. Furthermore, DMEM cultures were enriched with mouse cell contaminants, which completely overtook the culture in four cases (Fig. 1b, right panel).

Although cell differentiation was highly affected by the different culture conditions, it remained below 10% in most of the samples (8 out of 13 PPCs) consistent with the low fraction of terminally differentiated cells observed in primary RMS tumors[14,15] (Fig. 1b). Nevertheless, in a few PPCs such as SJRHB010463_X16 (PAX3-FOXO1 positive), SJRHB010468_X1C and SJRHB013757_X2 (both PAX7-FOXO1-positive), the percentage of MHC$^+$ cells exceeded 30% in some conditions, a phenomenon that was visible either after short-term culture or at later passages (Fig. 1b, Supplementary Fig. 2A) and that highlights the inherent myogenic differentiation potential of some RMS.

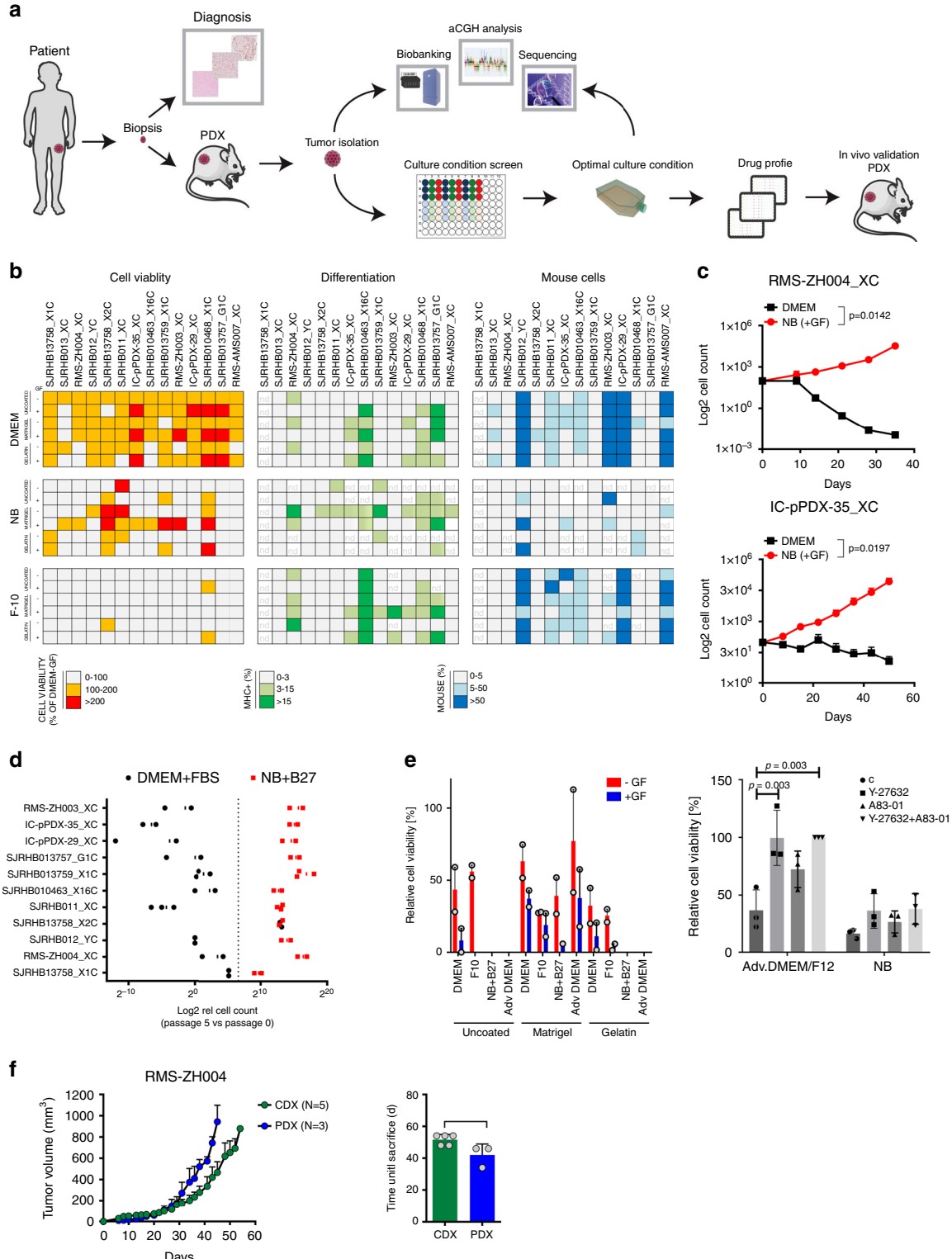

To further validate these findings, we next compared the long-term proliferation rate of PPCs in DMEM and NB medium. Counting of cell numbers of 6 FP-RMS and 5 FN-RMS cultures over a period of 1-2 months at every passage revealed that in 10 out of 11 cases (91%), NB-cultured cells continuously proliferated with an exponential growth rate, whereas cell counts in DMEM decreased over time starting already at passage 1 or 2 (Fig. 1c, d and Supplementary Fig. 2B). Only SJRHB13758_X2C cells could

easily be propagated in DMEM with a growth kinetic indistinguishable from the one in NB medium, reflecting the high take-rate and aggressiveness of this particular tumor observed in vivo (Fig. 1d and Supplementary Fig. 2B). Overall, this initial test series resulted in highly efficient (17/20 PDX) generation of PPCs and only in case of RMS-ZH002, RMS-ZH005 and RMS-AMS007 culture establishment failed. In these latter cases we additionally tested advanced DMEM/F12 medium containing

**Fig. 1 2D and 3D culture platform of PPCs. a** Workflow used to set up a RMS pre-clinical drug profiling platform. Briefly, PDXs were established from small RMS biopsies and first enzymatically dissociated to derive PPCs, followed by a culture condition screen (see text). DNA-sequencing and aCGH analysis were performed on both PDXs and paired PPCs to generate a sample collection with annotated genomic information. Finally, a high-throughput drug screen was conducted on each PPC cultivated under optimal culture conditions and top drug candidates were further validated in vivo on the original PDXs. **b** Culture optimization of indicated PPCs. Heat map depicting cell viability scores ($n = 2$–5 biological replicates), differentiation status ($n = 2$–3 biological replicates) and quantification of contamination with mouse stroma cells ($n = 2$–3 biological replicates) after short-term culture (1–3 weeks) of PPCs is shown. nd, not determined due to either low number of surviving cells or high fraction of cell loss during washing steps. **c** Growth curves of indicated PPCs cultured in DMEM on uncoated plates (black line) or in NB + GF on matrigel (red line). Cell number was normalized to day of seeding and is expressed in a log2 scale. (Mean ± range; $n = 2$ biological replicates). **d** Graph summarizing the proliferation rate of 11 PPCs under optimal and standard DMEM conditions. Data express the percentage of cells after 5 passages relative to day 0 in a log2 scale ($n = 2$–3 biological replicates). The dotted line indicates the cell number at day 0. **e**, left panel, Viability of the human cell fraction in PPCs from RMS-AMS007 PDXs cultured under indicated conditions (Mean ± range; $n = 2$ biological replicates). **e**, right panel, Viability of PPCs from RMS-AMS007 cultured in indicated media supplemented with ROCK- (Y-27632) and/or TGFβ- (A83-01) inhibitors (Mean ± sd; $n = 3$ biological replicates; two-way ANOVA with Tukey's multiple comparisons test). **f**, left panel, Tumor growth kinetics of RMS-ZH004 PDX (blue, $n = 3$ biological replicates) and passage 4 CDX (green, $n = 5$ biological replicates). (Mean ± sem). **f**, right panel, Histogram indicating the day (**d**) of sacrifice of individual CDX- and PDX-bearing mice (gray dots) when tumors reached ~1000 mm$^3$. (Mean ± sd; two-tailed paired $t$-test; NS, not significant;). Source data are provided as source data file.

ROCK and TGFβ pathway inhibitors in combination with the additives described above. Interestingly, these conditions allowed establishment PPCs in all these cases (Fig. 1e).

The optimized medium conditions also allowed propagation of PPCs as spheroids in a 3D culture system, with both size and cell numbers in spheroids increasing over time (Supplementary Fig. 3A, B). Immunohistochemical (IHC) characterization of 7 days old spheroids confirmed that the majority of cells was negative for cleaved Caspase-3 and expressed RMS markers (Myogenin and AP2β) (Supplementary Fig. 3A). Furthermore, we detected increased levels of the hypoxia marker Glut-1 in the center of large spheroids, indicating presence of an oxygen-gradient from the outside towards the inside of the spheres as expected (Supplementary Fig. 3A).

Finally, to evaluate whether in vitro culture in NB medium affects tumorigenicity of cells in vivo, we injected $5 \times 10^6$ RMS-ZH004 cells from in vitro cultures and dissociated parental PDX into NSG mice. In both models the engraftment rate was 100% and tumor growth as well as survival of tumor-bearing mice were very similar (Fig. 1f).

Altogether, these findings indicate that our platform can readily capture heterogeneity in culture requirements among different tumors and identified NB medium in combination with a matrix support as the optimal condition that outperforms conventional cell line protocols.

**Culture condition dependencies distinguish subgroups of RMS.** To better characterize our culture protocol, we aimed next to unravel the role of individual media constituents in more detail.

First, we assessed the contribution of GFs to cell viability and proliferation and calculated the ratio of cell viability in presence and absence of GFs (Fig. 2a). Remarkably, while GFs stimulated growth of FP-RMS, in FN-RMS they either were inert or dramatically impaired cell viability (SJRHB012_YC, SJRHB012_ZC, SJRHB011_XC and SJRHB011_YC), demonstrating differential GF demands among PPCs (Fig. 2a–c, Supplementary Figs. 1D and 4A). To identify the responsible GF, we evaluated the proliferation rate of GF-stimulated (SJRHB013759_X1C) and -inhibited (SJRHB011_YC, SJRHB012_YC and SJRHB012_ZC) PPCs under all possible permutations (Fig. 2c and Supplementary Fig. 4B). Notably, this analysis identified bFGF as the major player affecting proliferation (Fig. 2c and Supplementary Fig. 4B). Taken together, while these data confirm the well-known role of FGF signaling pathways for RMS cell proliferation, they also unravel a novel and unexpected anti-proliferative effect of bFGF in some FN-RMS.

Second, we aimed to determine the component(s) in the DMEM mixture which are detrimental to cell growth by measuring proliferation of four PPCs in both DMEM or NB-based media supplemented with either B-27, FBS or the combination of both (Fig. 2d and Supplementary 4C,D). This revealed that FBS limited cell growth in all PPC cultures irrespective of medium (Fig. 2d and Supplementary Fig. 4C, D).

Importantly, in DMEM cultures of two tumors (SJRHB012_YC and SJRHB012_ZC) we observed outgrowth of a few clones in DMEM medium (DMEM_clones), which then could be propagated to confluent monolayers within 1-2 months (Supplementary Fig. 4E, F). When transplanted into mice, both NB cultured cells and the DMEM clones were tumorigenic. Though the NB cultured cells and DMEM clones displayed similar growth kinetics with no significant differences in animal survival for SJRHB012_YC cells (Supplementary Fig. 4G), the DMEM clones from SJRHB012_ZC showed a more heterogeneous in vivo growth pattern and tumorigenic behavior than the corresponding NB cultured cells (Supplementary Fig. 4H). Thus, cells cultured in DMEM require a selection and/or adaptation process to re-gain proliferation which is not always homogenous in vivo.

Overall, this analysis identifies a difference in response to FGF pathway activation between FN- and FP-RMS and also implicates serum as an adverse component for PPC long-term proliferation (Fig. 2e).

**Molecular and histological characterization of PDXs and PPCs.** To assess whether our culture conditions exert a clonal selection pressure[16], we performed an in-depth molecular analysis of all matched PDX tumors and PPCs. First, we compared global DNA copy number alterations in matched PDX tumors and PPCs (passage<11) using array-CGH (aCGH). Overall, we detected a high level of concordance between PDXs and PPCs (Fig. 3a and Supplementary Fig. 5). Nevertheless, in accordance with previous studies, we also noticed some focal differences in DNA copy numbers between PDX and corresponding PPCs which were more pronounced in DMEM-derived clones (Fig. 3a).

Next, we analyzed the mutational landscape in PDXs and corresponding PPCs by exome sequencing. If available, we also compared data from PDX and PPC with patient germline data. This comparison confirmed that all somatic mutations identified in the PDX are also present in matched PPCs (Fig. 3b). The mutations found in our cohort reflect the common mutational spectrum of RMS. In FN-RMS we detected recurrent mutations in the RAS pathway, including *FGFR4*, *NRAS* and *HRAS* mutations, and *TP53*. Importantly, we also detected a close relationship

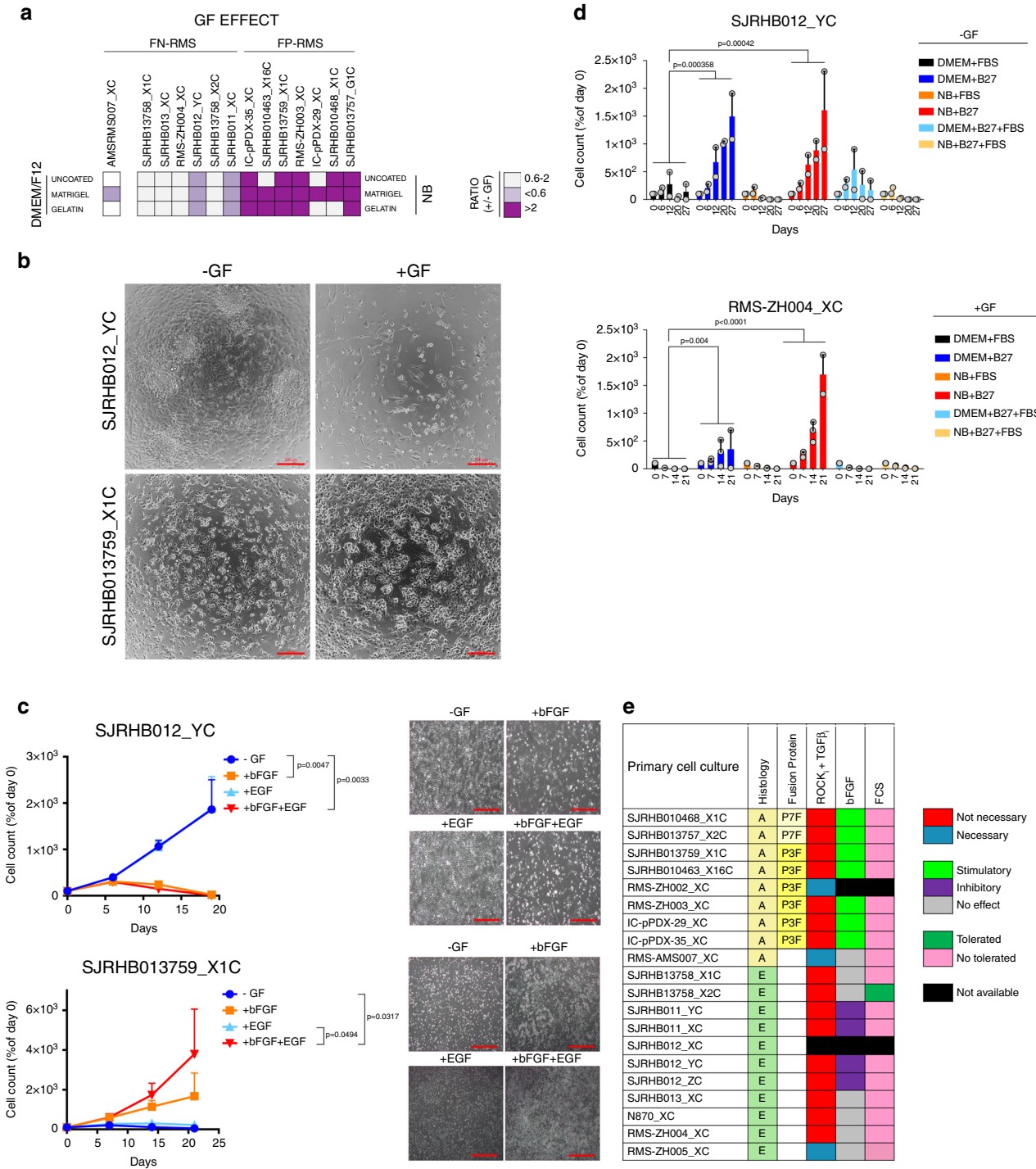

**Fig. 2 Culture condition dependencies distinguish subgroups of RMS. a** Heat map displaying the ratio of cell viability scores (as calculated from data shown in Fig. 1b and corrected for mouse contamination) in presence and absence of GFs (+/- GF) for FN-RMS (left) and FP-RMS (right) PPCs. Light and dark purple colors indicate a negative and positive influence of the GF on cell viability, respectively. White color represents a range of negligible effect. **b** Representative bright-field images of indicated cells at passage 1 cultivated with or without GF stimulation. Scale bar, 200 μm. **c** Assessment of GF dependency for long-term proliferation of GF-inhibited (SJRHB012_YC, upper panel) and GF-stimulated (SJRHB013759_X1C, lower panel) cells. Each data point is expressed as percentage of day 0. (Mean ± range; $n = 2$ biological replicates; two-way ANOVA with Tukey's multiple comparisons test). **c**, right panel, Representative light-microscopy images of the respective cells taken at passage 2. 100X magnification. **d** Histograms displaying cell counts over time for two PPC lines (RMS-ZH004_XC, SJRHB012_YC) cultivated under indicated conditions. -GF and +GF indicate the absence and presence of growth factors in the culture, respectively. (Mean ± range; RMS-ZH004_XC 0-14 days $n = 3$ biological replicates, RMS-ZH004_XC 21 days and SJRHB012_YC, $n = 2$ biological replicates; two-way ANOVA with Dunnett's multiple comparison test). **e** Heat map summarizing growth conditions found to be optimal for indicated PPCs. NB, NB medium supplemented with 2xB27, Adv.DMEM/F12, Adv.DMEM/F12 medium supplemented with 0.75xB27, Y-27632, A83-01 and N-acetylcysteine. Source data are provided as source data file.

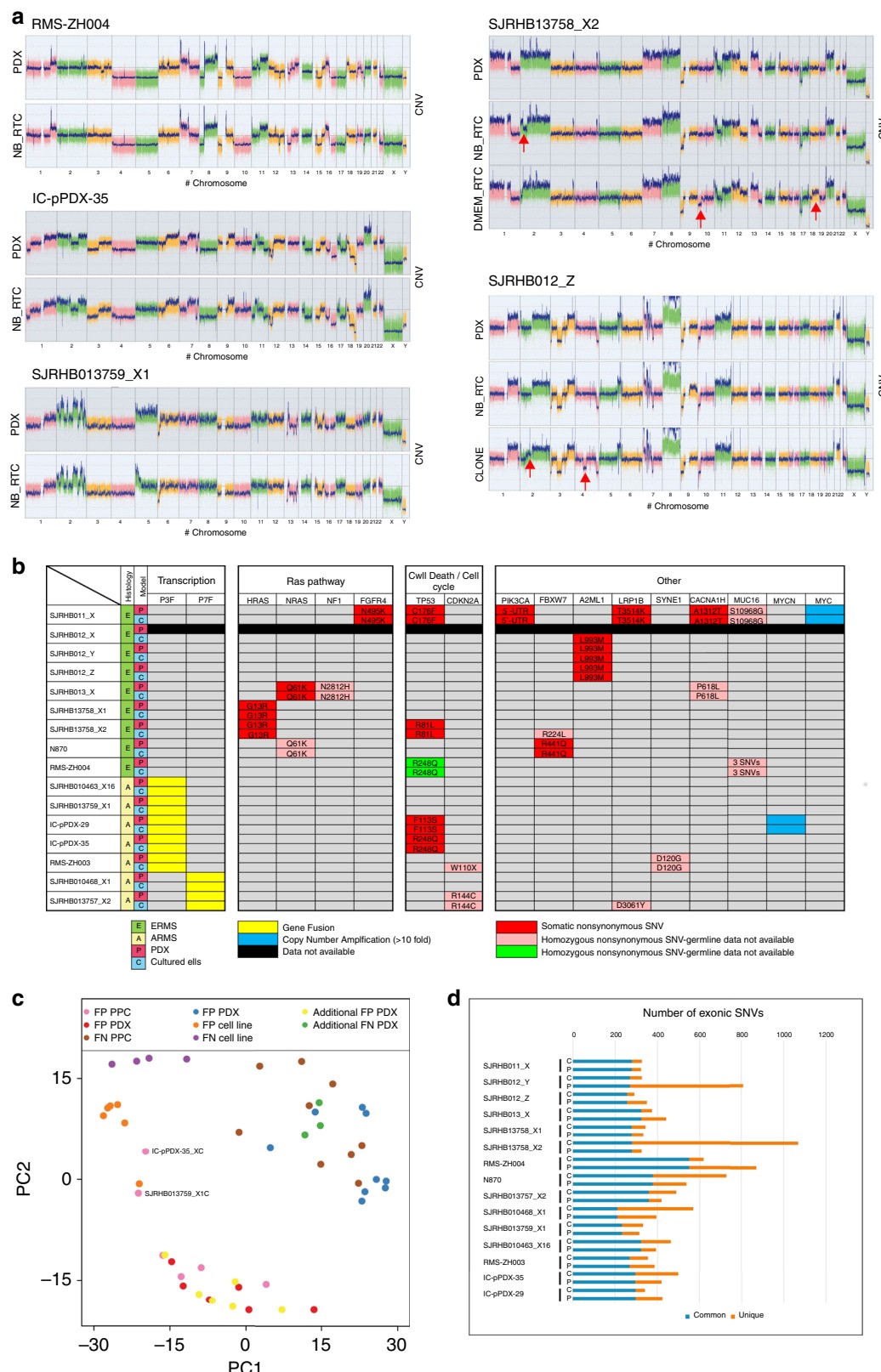

between the mutational status of *TP53* and the cellular response to idasanutlin, a MDM2-P53 interaction antagonist (Supplementary Fig. 6A), suggesting that increasing P53 protein levels in cells with non-mutant *TP53* remains an attractive therapeutic strategy. In FP-RMS the number of detected somatic SNVs was generally much lower. Expression of PAX3/7-FOXO1 fusion proteins was validated in all FP-RMS cultures by Western blot (Supplementary Fig. 6B). We then used the genewise target coverage of the exome seq data to identify focally amplified genes and matched the findings with the aCGH data. We detected amplifications of MYC (one FN-RMS) and MYCN (one FP-RMS) (Fig. 3b and Supplementary Table. 1).

**Fig. 3 Genomic and proteomic characterization of paired PDX and PPC samples. a** Copy-number variant (CNV) analysis of PDXs and corresponding PPCs (passage <11). Red arrows indicate detected copy number changes between PDX and PPC for samples where both DMEM- and NB-derived cultures were available. **b** Comparison of the mutational landscape of PDXs and corresponding PPCs. The table depicts all confirmed somatic SNVs in cases with available matched germline sequencing data. For samples without available germline data, comparison to dbSNP and 1000 g databases was used to identify potential non-synonymous SNVs in the same genes. Focal amplifications refer to genes with more than 10 gene copies. **c** Principal component analysis comparing DNA methylation profiles of RMS PDXs, PPCs and cell lines. DNA methylation data for 15 RMS PDX and PPC pairs, 9 additional PDXs and 10 cell lines were analyzed using the top 5000 most varied DNA methylation probes across all samples. Samples were indicated as color coded dots. Red and pink denote FP PDXs and corresponding PPCs respectively; blue and brown, FN PDX and corresponding PPCs; yellow and green, additional FP and FN PDXs; orange and purple, FP and FN cell lines. **d** Number of exonic SNVs in PDX and PPC model. P, PDX, C, PPC.

We also determined the stability of the models at both the epigenetic and genetic level. For the former we measured methylation profiles of 15 PDX/PPC pairs and used 8 common RMS cell lines (4 ARMS and 4 ERMS) as comparison. Principle component analysis (PCA) revealed that in 13 out of 15 cases PDXs and corresponding PPCs have similar methylation profiles and only two of the PDX/PPC pairs (SJRHB013759_X1 and IC-pPDX-35) showed a more divergent methylation pattern (Fig. 3c). Importantly, conventional cell lines clustered separately displaying much higher methylation levels at multiple sites.

To assess genetic stability we compared the number of exonic SNVs present in PDX and PPCs, respectively. Interestingly, in most pairs the number of SNVs was very similar (Fig. 3d). Only in SJRHB13758_X2C cells, we noticed a high number of unique SNVs that were not present in the parental PDX, indicative of genetic instability in the cultured cells.

To test whether histological RMS features are preserved in our models, we generated s.c. xenografts with passage 4-6 PPC cells (cell-derived xenografts; CDX) and compared their histological characteristics with the PDX and original patient tumors, if available. Tumor sections were assessed for cell and tissue morphology by haematoxylin and eosin (H&E) staining and for presence of cells with skeletal muscle differentiation by immunohistochemical detection of DESMIN and MYOGENIN. Impressively, both PDX and CDX show characteristic RMS architecture and a degree of MYOGENIN and DESMIN positivity, which is in line with published data showing that number of MYOGENIN positive cells discriminates ARMS from ERMS (Supplementary Fig. 7A, B).

Altogether, these findings showed that PPCs are epigenetically and genetically stable and faithfully recapitulate tumor histology when transplanted in vivo.

**In vitro compound screen with PPCs**. We next asked whether PPC cultures would represent a suitable pre-clinical model to unveil drug sensitivities in individual tumors. Therefore, we applied an in vitro proof-of-concept high-throughput screen employing a compound library containing 204 drugs which contained both Food and Drug administration (FDA)-approved drugs and small molecules in clinical development, covering a range of functional classes of targets, as well as standard chemotherapeutics used for RMS therapy (Supplementary Table 2). A panel of 17 PPCs (10 FN-RMS and 7 FP-RMS) and four established cell lines (FN-RMS cell lines RD and RH36 and FP-RMS cell lines Rh4 and Rh30) were cultured in 2D and treated for 72 h with a drug concentration of 500 nM. 63/204 (~30.9%) decreased cell viability by more than 40% in at least one sample, with a high concordance between biological replicates (Fig. 4a and Supplementary Fig. 8A). Unsupervised hierarchical clustering using the response data revealed that FP-RMS samples cluster together, while FN-RMS split into two branches (Supplementary Fig. 8B), reflecting both the different genetic landscape characterizing the two RMS subtypes as well as the larger heterogeneity of FN-RMS tumors[2]. At the level of individual drugs, we detected different response patterns. The minority of drugs

showed general toxicity including proteasome (4), HSP90 (2) and PI3K (2) inhibitors, as well as compounds interfering with the apoptotic machinery (YM155 and verdinexor), the dual ALK/IGF1R inhibitor AZD3463, and the mTOR inhibitor torin 2. Importantly however, most drugs showed more patient-specific activity patterns. Among them, both AKT and MEK inhibitors exhibited selective sensitivities (Fig. 4a). As proof-of-principle for reproducibility of our discovery drug platform we generated dose-response curves for several compounds anticipated to have either a general (verdinexor and YM155) or specific (ponatinib, dovitinib, ABT-263 and AZ20) effect and found agreement with the predicted cell viability scores (Supplementary Fig. 9A).

To test whether drug responses differ between 2D and 3D culture conditions, we compared the sensitivity of IC-pPDX-35_XC and SJRHB13758_X2C cells cultured as monolayer and as spheroids towards a selection of 12 drugs covering a broad range of half-maximal inhibitory concentration ($IC_{50}$) values. Importantly, $IC_{50}$ values of the different drugs were very similar in the two culture systems (Fig. 4B and Supplementary Fig. 9B), suggesting that drug response is not relevantly different in 2D and 3D cultures.

**Direct culture of RMS patient cells**. To move our platform towards a co-clinical situation, we next aimed to culture cells directly from patient tumor biopsies. For this, we isolated cells from a lymph node biopsy removed from FP-RMS patient RMS-ZH002 at the second relapse stage (Fig. 5a). Cultures of these cells established in NB medium supplemented with B27 and bFGF revealed to contain only tumor cells as determined by DNA fluorescent in-situ hybridization using probes detecting the FOXO1 translocation (50 out of 50 cells positive for two copies of rearranged FOXO1) (Fig. 5b, c). We therefore performed a drug profiling as described above, using a slightly larger drug library containing 250 drugs (Fig. 5d). Cells from the PDX generated from the first relapse were used as comparison. Among the identified effective drugs were several proteasome and HDAC inhibitors. Consequently, we validated the effects of the proteasome inhibitor Bortezomib and the inhibitor Panabinostat alone and in combination. The combinatorial treatment matrix revealed $IC_{50}$ values of 26 and 10 nM for Panobinostat and Bortezomib, respectively, and showed a synergistic effect of the two drugs at clinically relevant doses (Fig. 5e).

Taken together, this data demonstrates that the PPC drug profiling platform is compatible with cells directly isolated from patient tumors and allows detection of relevant patient-specific vulnerabilities in a reasonable time frame in a co-clinical setting.

**AKT is a potential therapeutic target in a subgroup of RMS**. Of particular interest among the responses detected by the drug profiling were the individual sensitivities towards AKT and MEK inhibitors. Correlation analysis further underscored the high degree of overlap among different AKTi and MEKi, suggesting on-target activities of these compounds (Supplementary Fig. 9C).

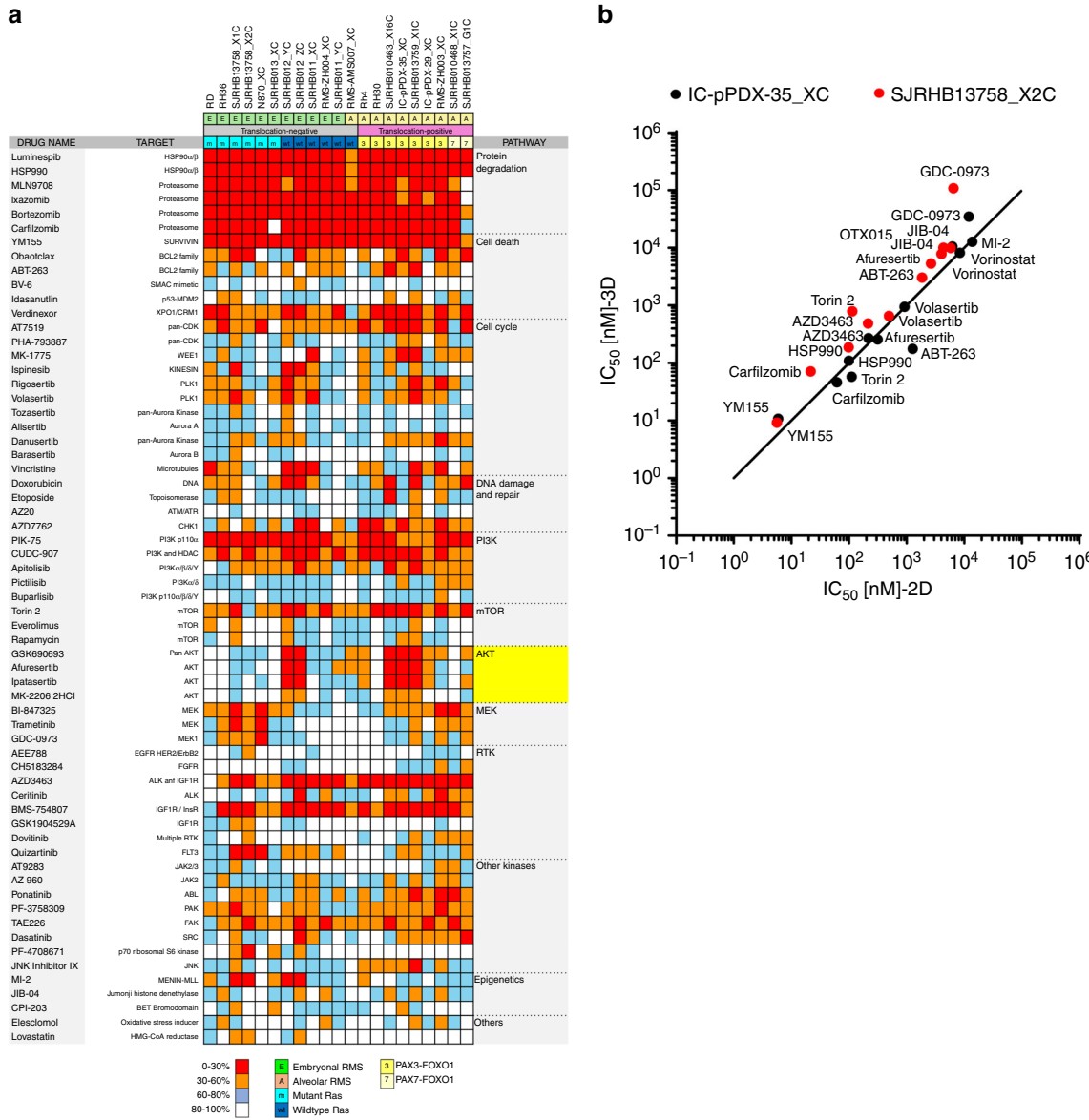

**Fig. 4 2D and 3D in vitro drug profiling of PPCs. a** Heat map depicting the activity of indicated agents used at a final concentration of 500 nM across the panel of RMS cells. Histological subtype, fusion status and *RAS* mutation status are shown above each column. Each color in the heat map indicates the percentage of remaining cell viability after treatment relative to vehicle control: 0–30% (red), 30–60% (orange), 60–79% (light blue) and >79% (white) (Mean; $n = 2$ biological replicates). **b** Correlation plot comparing $IC_{50}$ values for indicated drugs determined in 2D and 3D cultures of IC-pPDX-35 and SJRHB13758_X2C cells. (Mean; $n = 2$–3 biological replicates). Source data are provided as source data file.

Remarkably, we could discriminate three subgroups in our PPC panel: Two groups are responsive to either AKT or ERK inhibition and a third group is refractory to both (Fig. 6a). Interestingly, drug responses could not be predicted by the genetic profile, and sensitive cases included both FP- and FN-RMS. In contrast, inhibition of downstream mTOR by torin-2 showed a more uniform toxicity (Supplementary Fig. 9D).

In order to study the interplay between the two pathways, we next treated eight PPC cultures with a combination matrix of afuresertib and trametinib to block both AKT and MEK, respectively. Calculation of BLISS scores to evaluate synergism revealed a synergistic behavior of the two drugs in all but one of the cases, which was especially evident for the *RAS* mutant ERMS, with BLISS scores of 10-20 (Fig. 6b and Supplementary Fig. 10A). $IC_{50}$ values for trametinib alone differed nearly 2 logs among four *RAS* mutant cases (Fig. 6c), with more sensitive cases responding uniformly with cell death, while in the most resistant case

(SJRHB13758_X2C) up to 30 percent of the cells differentiated (Supplementary Fig. 10B, C). $IC_{50}$ of the most sensitive cases ($IC_{50} < 10$ nM) were in the range of therapeutically applicable concentrations. The same was the case for afuresertib ($IC_{50} < 281$ nM) and a second AKT inhibitor (GSK690693) for the most sensitive PPCs (Fig. 6d). Western blot analysis in sensitive (IC-pPDX-35_XC) and resistant (RMS-ZH004_XC) cases confirmed that afuresertib selectively inhibited phosphorylation of S6 Ribosomal Protein and released a negative feedback loop on AKT phosphorylation similar to previous reports[17] (Fig. 6e and Supplementary Fig. 11A). We also observed a strong increase in number of floating cells as well as cleaved Caspase 3/PARP specifically in sensitive cells (Supplementary Fig. 11B), indicating that these cells die upon treatment.

Next, to explore the reliability of our pre-clinical in vitro platform for prioritizing compounds, we tested the response to afuresertib in vivo. For this, PDX from a sensitive (IC-pPDX-35_X) and a

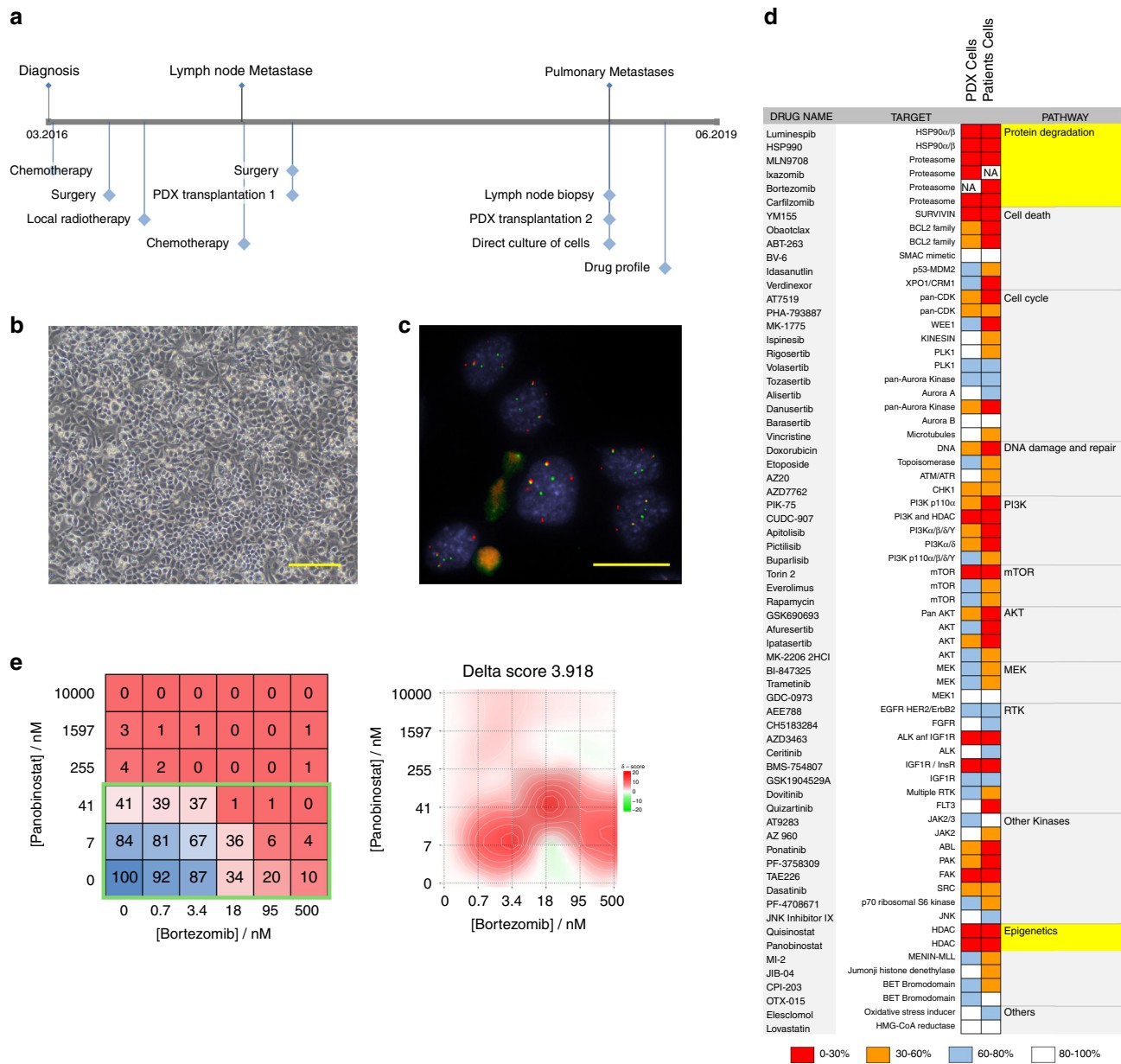

**Fig. 5 Direct culture of RMS patient cells for drug profiling and therapy guidance. a** Scheme depicting the clinical course of patient RMS-ZH002. **b** Bright-field image of cells isolated from the lymph node biopsy of patient RMS-ZH002. Scale bar 400 μm. The displayed image is representative of $n = 1$ individual cell isolation. **c** Fluorescence in situ hybridization detection of the *FOXO1* gene using probes binding to genomic sites N- (red) and C-terminal (green) of the gene. The displayed image is representative of $n = 1$ fluorescence in situ hybridization experiment. **d** Heat map depicting the activity of indicated agents used at a final concentration of 500 nM on cells isolated from PDX RMS-ZH002 and on cells isolated from the lymph node biopsy. Each color in the heat map indicates the percentage of remaining cell viability after treatment relative to vehicle control: 0–30% (red), 30–60% (orange), 60–79% (light blue) and >79% (white). **e**, left panel Heat map depicting effect of panobinostat and bortezomib alone and in combinations on viability of RMS-ZH002 patient cells isolated from the lymph node biopsy (Mean; $n = 2$ biological replicates). Clinically applicable concentrations of the two drugs are framed with a green box. Right panel, Heat map depicting synergism between panobinostat and bortezomib using zero interaction potency model analysis. Source data are provided as source data file.

resistant (RMS-ZH004_X) patient were treated with afuresertib for at least three consecutive treatment cycles. Strikingly, growth of the RMS-ZH004_X tumor (Fig. 6f, lower panel) was not altered unlike the strong response observed for IC-pPDX-35_X PDX, which showed stable disease within the first three weeks of treatment (Fig. 6f, upper panel). Consequently, sensitive but not resistant tumor-bearing mice lived significantly longer than control groups with a difference in median survival of ~19 days (Fig. 6f, right panels). Western Blot analysis using tumor extracts after short-term in vivo treatment (7 days) proved on-target activity of the drug

(Supplementary Fig. 11C) whereas the levels of phosphorylated S6 ribosomal protein consistently increased upon prolonged therapy, consistent with the late-stage tumor re-growth following the strong initial response to the drug (Supplementary Fig. 10D).

In order to further validate the combinatorial effects of AKT and MEK inhibition in vivo, we treated IC-pPDX-35_X tumors with the combination of afuresertib and trametinib using low doses of both drugs. Similar to the in vitro results, we also detected a synergistic behavior of the two drugs in vivo (Fig. 6g).

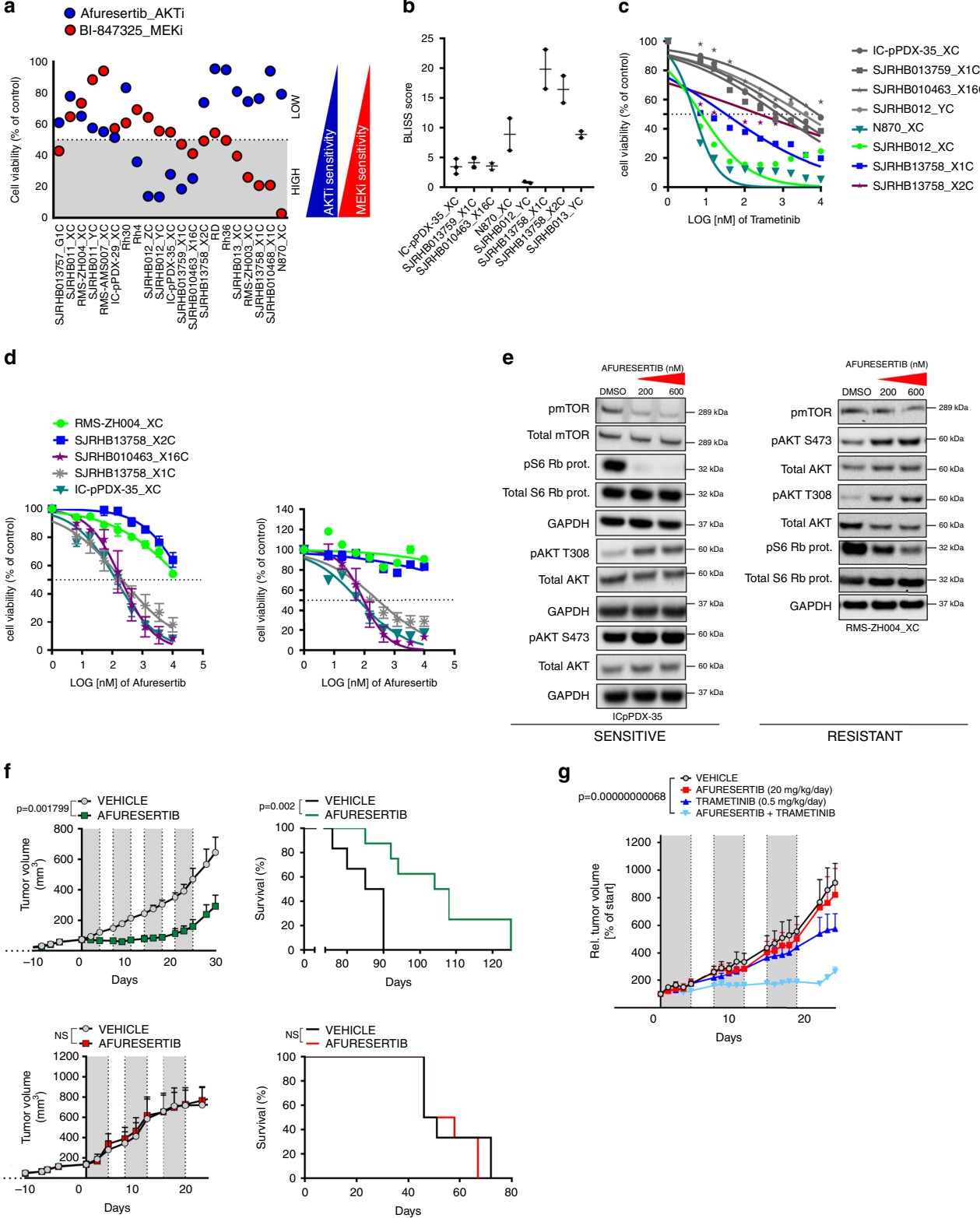

In conclusion, we describe a functional and reliable proof-of-concept drug profiling platform that allowed to identify a novel subgroup of AKTi sensitive RMS that could not be revealed by genomic-based predictions. These findings highlight the suitability of our platform to identify functional signaling dependencies in RMS.

## Discussion

In the present study we have established a biobank of PDX-derived primary rhabdomyosarcoma cultures using optimized serum-free culture methods that preserves niche factor requirements, patient-specific genomic alterations, proliferative capacity and tumorigenic ability.

**Fig. 6 In vitro and in vivo validation of afuresertib and trametinib for RMS treatment. a** Comparative analysis of afuresertib (AKTi, blue) and BI-847325 (MEKi, red) effects on cell viability. Data were computed from the drug screen experiments. Gray and white areas indicate the range below and above 50 percent inhibition of cell viability, respectively, which were chosen as range of high and low drug response (Mean; $n = 2$ biological replicates). **b** BLISS synergy scores from the combination treatment of indicated cells treated with a combination matrix of afuresertib and trametinib (Mean with range; $n = 2$ biological replicates). Scores were determined with the synergyfinder webtool. **c** Cell viability of indicated PPCs treated for 72 h with increasing concentrations of the MEK inhibitor trametinib. *RAS* mutant samples are depicted in colors, *RAS* wild type samples are in gray. (Mean; $n = 2$ biological replicates). **d** Cell viability of indicated PPCs treated for 72 h with increasing concentrations of the AKT inhibitors afuresertib (left panel) and GSK690693 (right panel). (RMS-ZH004_XC and IC-pPDX-35_XC, mean ± range, $n = 2$ biological replicates; SJRHB13758_X2C, SJRHB010463_X16C and SJRHB013759_X1C, mean ± sd, $n = 3$ biological replicates). **e** Western blot analysis showing expression and phosporylation status of indicated proteins from PPCs treated for 2.5 h with afuresertib at 200 nM and 600 nM or DMSO as control. Both an afuresertib-sensitive (IC-pPDX-35_XC) and -resistant (RMS-ZH004_XC) case is presented. GAPDH was used as loading control. The displayed blots are representative of $n = 2$ independent experiments. **f**, upper left panel Tumor growth of IC-pPDX-35 PDX (sensitive) treated with 100 mg/kg per day of afuresertib (green line) or vehicle control (black line). Gray color depicts treatment time frame (5 days) for four consecutive treatment cycles. (Mean ± sem; $n = 6$ mice; two-way ANOVA). **d**, Upper right panel, Kaplan–Mayer curves of mice from left panel. (Log rank (Mantel-Cox) test). **f**, Lower left panel, Tumor growth of RMS-ZH004 PDX (resistant) following the same treatment schedule as in the upper panel but for three consecutive cycles. (Mean ± sem; $n = 6$ mice; two-way ANOVA; NS, not significant). (Lower right panel) Survival curve of mice from left panel. (Log rank (Mantel-Cox) test; NS, not significant). **g** Tumor growth of IC-pPDX-35_XC PDX treated with vehicle controls (black line), 20 mg/kg per day of afuresertib (red line), 0.5 mg/kg per day of trametinib (dark blue line) or the combination of the two drugs (light blue line). Gray color depicts treatment time frame (5 days) for three consecutive treatment cycles. (Mean ± sem; control and afuresertib, $n = 5$ mice; trametinib, $n = 4$ mice; combination, $n = 3$ mice; two-way ANOVA with Tukey's multiple comparisons test). Source data are provided as source data file.

Our culture screen revealed a general difficulty to propagate primary RMS cells in serum-containing media, consistent with the low rate of success to establish cancer cell lines in the past with standard culture protocols[18]. Similar results have been documented for primary glioblastoma, neuroblastoma and ovarian cancer cells, suggesting that this might be a more general phenomenon[19–21]. We speculate that the multitude of components of the serum might not properly recapitulate the fluid that cells are exposed to in their natural milieu[22]. Accordingly, two out of three serum-tolerant cultures in our cohort were associated with a clonal outgrowth reminiscent of cell cultures undergoing crisis before re-gaining a proliferative status[19,23]. The selection pressure imposed by this culture condition resulted in the emergence of additional genetic aberrations and, in one case, even limited the tumorigenic potential of the cultured cells. This occurrence resembles the loss of tumorigenicity observed in primary glioblastoma cells exposed to serum[19]. While future experiments would be needed to unravel individual inhibitory factors in serum and clarify the mechanisms of stress-induced growth arrest, recent studies have suggested that both senescence and differentiation can be observed[20,24].

However, apart from these similarities in cell growth, a main finding of our study is the heterogeneity among RMS PPCs at different levels including inherent differentiation potential, cell culture condition demands and drug response. Regarding differentiation potential, we found that especially the two PAX7-FOXO1-positive PPCs, but to a lesser extent also some other PPCs, exhibited a high myogenic differentiation potential when cultured under appropriate conditions. This suggests that pro-differentiation strategies might be a promising therapeutic route in these tumors. Our data also strongly suggest that specific culture parameters have to be developed for different sarcoma types or even subgroups within these entities. In case of two FP-RMS, one FN-RMS and multiple osteosarcoma, none of the 18 different conditions initially tested supported growth of cells in vitro. Interestingly, in the three RMS cases, TGFβ pathway and ROCK inhibition allowed in vitro propagation of the cells. Originally, ROCK inhibitors have been found to promote survival of stem cells in in vitro cultures[25] and have become part of standard cell culture protocols for stem cells as well as cells from ectodermal origin including normal epithelia and carcinoma[26–28]. At this point it remains unclear whether culture dependencies reflect cell/lineage of origin of tumors, a question that warrants further investigations.

A second striking example of heterogeneity in culture conditions was the influence of bFGF on proliferation of PPCs. While it strongly induced proliferation of FP-RMS as anticipated, unexpectedly half of the FN-RMS were strongly inhibited. A similar observation has been described for Ewing cell lines where it involved cell death after sustained activation of p38MAPK[29,30]. In RMS, importance of FGFR signaling for tumorigenicity has been described in several previous studies. In FP-RMS, FGFR4 is a target gene of the fusion proteins and therefore highly overexpressed, while in a subset of FN-RMS the same receptor is activated by mutation and/or amplification[31–33]. Especially mutant FGFR4 might represent a potential therapeutic target[34]. Strikingly, two of the FN-RMS samples inhibited in our cohort (diagnostic and relapse samples from the same patient) expressed amutant FGFR4, suggesting that the signaling downstream of the FGFRs in RMS cells is more complex than previously appreciated and might have anti-proliferative effects under some circumstances.

A further level of heterogeneity was detected when analyzing drug response. Only few drugs had a general toxic effect among PPCs, including different proteasome inhibitors or the survivin inhibitor YM155. For many other classes of agents however, we detected heterogeneous patterns of responses across our set of PPCs. This resembles very well the heterogeneity of responses seen in the clinics towards first-line therapies. Importantly, we could demonstrate that drug sensitivities obtained from the primary 2D cultures accurately predict in vivo response. In this context it is not surprising that we could not detect major differences in drug response between monolayer and spheroid cultures in vitro. It seems that the bulk of the cells responds similarly in all three model systems (in vivo, in vitro in 2D and 3D). This might be surprising taking into account the large efforts that are undertaken to develop 3D model systems for drug screenings, especially for carcinoma. However, they are in agreement with similar 2D screening approaches, which have resulted in identification of cancer specific vulnerabilities in adult cancers[8]. Additional work will be necessary to evaluate whether 3D models offer an additional benefit measuring drug responses in specific cellular subpopulations, such as the tumor promoting compartment in FN-RMS tumors[35–37]. Taken together, heterogeneity at

different levels strongly argues for a functional precision medicine approach and might be taken into account in screening programs using a fixed number of models[38]. Importantly, experimental-designs using one mouse per tumor have been proposed as approach to test a large number of different tumors in vivo in parallel[39].

Among the most interesting drug sensitivities were extra-ordinary patient-specific sensitivities towards AKTi and MEKi that could not be predicted by genomic data. AKTi are currently developed in phase I/II clinical trials for adult malignancies[40]. Despite some evidence suggesting that AKT phosphorylation (AKT Ser473) predicts poor overall survival in RMS, little attention has been drawn on blockade of the AKT/mTOR axis so far[41]. Our search for potential biomarkers of these drug sensi-tivities failed to identify a close relationship between genomic features and drug responses. Instead, our findings indicate that in vitro drug profiling on PPCs is predictive of the response of the parental PDX and therefore is a suitable approach to pre-select RMS sub-groups who may benefit from anti-AKT therapies.

Finally, we also demonstrated that the optimized culture con-ditions allow culture of cells directly isolated from patient tumors. Circumventing the PDX generation has the advantage that the procedure is potentially significantly accelerated. Time until drug profiling is of vital importance, especially in the relapse situation when standard therapies are not effective anymore. Amount and quality of material available for isolation of the cells is an important parameter for this approach.

Overall, our findings provide important groundwork for a clinically used drug screening platform for sarcoma with the aim to offer drug response profiling as tool for therapeutic decisions for individual patients in the clinics.

## Methods

**Patient consent and sample collection**. All patients gave written informed consent at the participating institutions. This includes general informed consent from TATA Memorial Hospital and patient consents from University Children's Hospital Zurich, Emma Children's Hospital Amsterdam and Institut Curie Paris. Orthotopic PDX models from St Jude Children's Hospital were obtained through Childhood Solid Tumor Network[42], for which patient consent has been published previously[6]. Fresh biopsy material from human tumors was frozen in DMEM supplemented with 10% FBS and 10% DMSO at −80 °C in a cell freezing container before shipment to the the site of transplantation took place.

**PDX transplantation**. For tumor amplification, pieces with a size of ~10–30 mm$^3$ from fresh human tumor biopsis or established PDX tumors were transplanted subcutaneously into the flank of 6–10 weeks old, sex-matched NOD scid gamma (NSG) mice. Engraftment of tumors was monitored by tumor size measurements three times a week using a caliper. Tumors were isolated when they reached a size of ~1000 mm$^3$.

For in vivo drug treatment, a single cell suspension of dissociated PDX tumors containing $0.7–5 \times 10^6$ cells was injected subcutaneously into the flank of sex matched, 6–10 weeks old NSG mice. A description of the PDX lines used in this study can be found in Supplementary Fig. 1A.

**Isolation of PDX cells**. PDX tumors freshly isolated from mice were first minced using two scalpels and then incubated in a dissociation buffer composed of 200 µg/ ml liberase DH (Roche, 5401054001), 0.1 mg/ml DNase I (StemCell, #07900) and 1 mM MgCl$_2$ in 1x HBSS buffer (Sigma, H6648) for 30–60 min at 37 °C. When the tumor cell suspension was pipettable through a 1 ml tip, digestion was stopped with serum containing medium and remaining tumor pieces were removed by filtration through a 70µm cell strainer. Cell suspensions were then washed with PBS and either directly used for cell culture or frozen in freezing medium (CryoStor CS10, StemCell, #07930) for future use.

**Culture of PDX cells**. Media used for culture of PDX cells included DMEM (Sigma, D5671) or F10 (Life technologies, 11550-043) both supplemented with 10% heat inactivated FBS (Thermofisher Scientific), NB medium (Thermofisher Scien-tific, 21103049), supplemented with 2xB-27 (Thermofisher Scientific, 17504044), and Advanced DMEM/F12 (Thermofisher Scientific,12634010), supplemented with 1× or 2× (for drug screening) B-27, 1.25 mM N-acetyl-L-cysteine (Sigma-Aldrich, A9165), 5 µM A83-01 (Tocris Bioscience, 2939), and 10 µM Y-27632 (Abmole Bioscience, M1817). All media were further supplemented with 100 U/ml

penicillin/streptomycin and 2 mM glutamax (Life technologies, 35050-061). When indicated, media were further supplemented with 20 ng/ml bFGF (PeproTech, AF-100-18B) and 20 ng/ml EGF (PeproTech, AF-100-15). Three times per week, 75% of medium was replaced with fresh one. When reaching confluency, cells were detached using Accutase (Sigma-Aldrich, A6964) and splitted in a ratio of 1:2 to 1:3.

For Matrigel-coating, Matrigel (Corning, 354234) was diluted 1:10 in NB medium and left on the dish for 30–60 min at RT. Before cell-plating, excess Matrigel solution was removed. For Gelatin-coating, a 2% solution of Gelatin (Sigma-Aldrich, G9391) in water was left on the dish for 2 h at 37 °C. The solution was then removed and the plates were dried at RT for 30–60 min.

For spheroid culture, 5000 (SJRHB13758_X2C) or 10'000 (IC-pPDX-35_XC) cells were plated per 96 well in round bottom cell repellent plates (Greiner Bio-One, 650970) in NB medium supplemented with Matrigel (1:50). Half of the medium was changed three times a week.

**Culture condition optimization**. 30'000 cells freshly isolated from PDX tumors were plated per 96 well in the different test media. When cells reached confluency in one well, representative phase contrast pictures were taken from each condition, before cell viability was measured by WST-1 assay. The same plate was then used for immunofluorescence detection of myosin heavy chain (MHC) and quantifi-cation of mouse stroma cells by DAPI staining.

**Quantification of mouse cells**. For quantification of mouse stroma cells con-taminating PPC cultures, cells isolated from PDX and grown under different conditions in 96 well plates were stained with DAPI. Mouse cells were then identified on microscopic images by the punctate DAPI-staining pattern of their nuclei. Relative numbers of human and mouse cells were determined on micro-scopic images containing at least 200 cells of possible.

**Immunofluorescence**. Immunofluorescence was performed in 96 well plates. Cells were first fixed with 4% paraformaldehyde for 15 min, followed by quenching with 0.1 M glycine in PBS for 5 min and permeabilization with 0.1% Triton-X 100 in PBS for 15 min. After blocking with 4% horse serum in 0.1 % Triton-X 100 in PBS, cells were incubated with primary antibody diluted in the same buffer at 4 °C over night in a humid chamber. After three washings with PBS for 5 min, cells were incubated with secondary antibody diluted in PBS with 4% horse serum for 1 h. A monoclonal anti-MHC antibody (MF 20; Developmental Studies Hybridoma Bank, Iowa) (1:500) and an Alexa 594-labeled donkey anti-mouse secondary Ab (Ther-mofisher Scientific, A11032) (1:500) were used. For nuclear staining, cells were covered with PBS containing 10 µg/ml DAPI and analyzed by fluorescence microscopy.

**Immunohistochemical staining of spheroids and tumor sections**. 2 µm sections of formalin-fixed, paraffin-embedded (FFPE) spheroids or tumor tissues were stained on the Bond automated staining system (Leica). For epitope retrievel, slides were first either enzymatically predigested with Ventana protease 1 for 10 min (Desmin) or heated in Epitope retrieval solution from Leica Bond system (PH9) to 100 °C for 20 min (Myogenin), 30 min (AP2b and Glu-1) or 45 min (cleaved Caspase-3). The sections were then incubated for 30 min with primary antibodies against Desmin (Novocastra Laboratories Ltd, NCL-L-DES-DERII) (1:200), Myo-genin (Cell Marque Lifescreen 296M-14Ltd) (1:100), AP2b (Santa Cruz, H-87) (1:400), cleaved Caspase-3 (Cell Signaling Technology, #9661) (1:500) and Glut-1 (Cell Marque Lifescreen Ltd., CMC35511020) (1:300). Visualization of the anti-bodies was performed with a Bond refine detection system, Leica. All sections were counterstained with haematoxylin.

**Measurement of long-term proliferation of cells**. Cells were cultured in parallel under optimal conditions as determined by condition screening and in conven-tional DMEM supplemented with FBS. Cell numbers were counted at each passage.

**Drug profiling**. For drug profiling, 3000–20,000 PPCs in NB medium in most cases supplemented with bFGF and EGF were plated per 384 well coated with Matrigel or Gelatin. Only the GF-sensitive PPC (SJRHB012_YC, SJRHB012_ZC, SJRHB011_XC and SJRHB011_YC) were cultured in absence of bFGF and EGF. For drug screen in cell lines, 2000-4000 cells were seeded in NB medium supple-mented with bFGF and EGF and in absence of any substratum. The next day, medium was changed and cells were incubated with a drug library containing 204 different drugs (Selleckchem) in a concentration of 500 nM in duplicate wells for 72 h. 12 wells treated with DMSO on each plate served as controls. Cell viability was then determined by WST-1 assay (Roche, 11644807001). A list of compounds employed in this study is provided in Supplementary table 2.

**Drug response curves**. For IC$_{50}$ determination in 2D cultures, 3000–20,000 PDX cells were plated in NB medium supplemented with bFGF and EGF per 384 well (unless otherwise specified) coated with Matrigel or Gelatin. The next day, medium was changed and cells were incubated with drug in a logarithmic concentration range between 0.006–10 µM using a digital dispenser (HP D300). After 72 h cell

viability was measured by WST-1 assay or CellTiter-Glo 3D assay (Promega, G9681) when compared to 3D cultures. For drug test in 3D cultures, cells were cultured as spheroids in 96 well plates for 7 days as described above. Spheroids were then treated and incubated as above and cell viability was measured using the CellTiter-Glo3D assay.

**Caspase 3/7-activity assay**. Cells were seeded in white 384-well plates with clear bottom (Greiner Bio-One, #781098). Caspase activity was determined at indicated time points by Caspase-Glo 3/7 Assay (Promega, #G8093) according to the manufacturer's instructions. Luminescence was measured using the multidetection microplate reader Synergy HT (Bio-Tek Instruments).

**High throughput microscopy**. An Operetta high content screening system was used for high throughput analysis of cells immunofluorescently stained for myosin heavy chain. Identification and quantification of myosin heavy chain positive cells was performed with the Harmony software.

**In vivo drug treatment**. Tumors were established from dissociated PDX and tumor harboring mice were randomized into treatment and control cohorts of 6 animals when the tumor average size reached about 100 mm³ (single afuresertib experiment) or individually distributed into different treatment groups when tumors reached about 100 mm³ (combination experiment). For afuresertib treatment, drug (100 or 20 mg/kg) and vehicle were administered by oral gavage 5 times a week. Afuresertib (Selleck, S7521) was dissolved in 20% PEG-400 (Lipoid), 1% DMSO and 79% H₂O. For trametinib (ApexBio, A3018) treatment, drug (0.5 mg/kg) was dissolved in 4% DMSO/ corn oil and administered by i.p. injection five times a week. Tumor size was measured three times a week using a caliper and mouse weight was measured twice a week. No mice needed to be euthanized due to severe body weight loss (>20% than baseline).

**Western blot**. Cell lysates for Western Blots were generated using RIPA buffer (50 mM Tris-Cl (pH 7.5), 150 mM NaCl, 1% NP-40, 0.5% Na-deoxycholate, 1 mM EGTA, 0.1% SDS, 50 mM NaF, 10 mM sodium β-glycerolphosphate, 5 mM sodium pyrophosphate, 1 mM sodium orthovanadate), and supplemented with Complete Mini Protease Inhibitor cocktail (Sigma-Aldrich, # 11697498001). Proteins were separated using NuPAGE™ Novex™ 4-12% Bis-Tris gels (ThermoFisher) and transferred to nitrocellulose membranes (GE Healthcare Life Sciences) by wet-blotting. Membranes were then blocked with 5% milk or BSA in TBS/0.05% tween for 20 min, followed by incubation with the primary antibody overnight at 4 °C. After three washing steps with TBS-0.05% tween for 5 min, membranes were incubated with a horseradish peroxidase-linked secondary antibody for 1 h at RT. After three additional washing steps with TBS/0.05% tween for 5 min and one final wash step with PBS for 1 min, proteins were detected by chemiluminescence using either the Pierce™ ECL or the Supersignal Western blotting reagent (both ThermoFisher) and a ChemiDoc MP imager (BioRad). Antibodies used included the following ones from Cell Signaling: anti-phospho-mTOR (#2971), anti-mTOR (#2983), anti-phospho-S6 ribosomal protein (#2211), anti-S6 ribosomal protein (#2217), anti-phospho-AKT Thr308 (#9275), anti-phospho-Akt Ser473 (#9271), anti-AKT (#9272), anti-GAPDH (#2118), anti-PARP (#9542), anti-cleaved CAS-PASE 3 (#9664). Antibodies against P53 was purchased from ThermoFisher Scientific (#AHO0152) and against FOXO1 from Santa Cruz Biotechnology (sc-11350). All primary antibodies were diluted 1:1000 in milk. For secondary antibodies, HRP-linked anti-mouse IgG (Cell Signaling, #7076) and HRP-linked anti-rabbit IgG (Cell Signaling, #7074) were used at 1:5000 dilution in milk.

**DNA copy number analysis**. Genomic DNA was extracted from PDX pieces or cultured cells using the DNAse® Blood&Tissue Kit (Qiagen, #69506) following the manufacturer's instructions. The aCGH assay was performed using the CytoScan™ HD Array Kit according to the manufacturer's protocol (Affymetrix, Thermo Fisher Scientific, MA, USA). The raw data of each single sample was analyzed with the Chromosome Analysis Suite (ChAS) software (Version 3.1.1.27, Affymetrix).

**Exome sequencing and analysis**. DNA was isolated from PDX and PPCs using the Qiagen DNeasy Kit. For exome sequencing, exome enrichment was performed using the Agilent SureSelectXT Human All Exon V6 kit. Paired-end sequencing was performed using the Illumina system. The sequences were demultiplexed using bcl2fasta in the illumina CASAVA pipeline to get fastq files. For processing of the data, the whole exome sequencing pipeline web tool was used[43]. This included the bwa tool (0.7.10) for alignment of the sequences with the human genome, Picard (1.119) for duplicate removal and GATK (2.8-1) with the GATK algorithm for calling of SNVs and insertion/deletions. Variants were further filtered using the following criteria: Coverage ≥10, ambiguous mappings per variant ≤5, Phred-scaled consensus quality ≥50, variant confidence quality ≥1.5, strand bias Fisher exact test ≤60, not present in dbSNP, not present in 1000 g (minor allele frequency <0.01).

Genewise target coverage data was used to identify amplified genes, using a threshold of 10 fold[43].

**Genome-wide DNA-methylation analysis**. Genomic DNA from eight PDX/PPC pairs was treated with bisulfite (EZ-96 DNA Methylation Kit [Zymo Research])[44], and then DNA methylation was analyzed on the Infinium MethylationEPIC (EPIC) BeadChip (Illumina). In addition, DNA methylation data from 12 RMS PDXs and 10 conventional RMS cell lines previously generated on the Infinium HumanMethylation450 (HM450) BeadChip were included in this analysis. Raw IDAT files from both sources were processed and normalized using the noob method in the minfi package[45,46]. Only probes that overlapped between EPIC and HM450 were studied in this analysis. Probes with detection $P$-value >0.01 in at least one sample, probes located on X and Y chromosome, non-CpG probes, probes containing a SNP at the single-base extension or CpG site, probes with genetic variants overlapping the body of the probes and probes identified as cross-hybridizing were discarded[47]. The β-value was computed as the measure of methylation, ranging from 0 (completely unmethylated) to 1.0 (complete methylated methylated). Hierarchical clustering and principal component analyses were performed as described[44]. Heatmaps were generated using the heatmap.plus packages in R. Principal component analysis was applied using prcomp function in stats package.

**Fluorescence in situ hybridization**. Cells cultured on chamber slides were fixed with Methanol-Acetic acid (3:1) for 10 min at −20 °C. FISH evaluation of FOXO1 rearrangement was performed with the FOXO1 break-apart probe (Cytocell, Cambridge, U.K.) according to the instructions of the manufacturer. Briefly, slide and probe were co-denatured at 80 °C for 2 min, followed by overnight hybridization at 37 °C and humid conditions using a Leica ThermoBrite (Biosystems Switzerland AG, Muttenz, Switzerland). Then, the slides were washed in a 0.4x saline sodium citrate (SSC) / 0.3% (vol/vol) IGEPAL solution (Sigma-Aldrich) for 2 min at 72 °C, followed by 1 min in a 2× SSC / 0.1 % (vol/vol) IGEPAL solution at room temperature. The slides were air dried, and the nuclei were counterstained with Vectashield Mounting Medium containing DAPI (REACTOLAB S.A., Servion, Switzerland). Microscopic images were acquired using the Axio Imager.Z2 microscope (Carl Zeiss AG, Feldbach, Switzerland) and 50 nuclei were analyzed using the Isis software (MetaSystems Hard & Software GmbH, Altlussheim, Germany).

**Unsupervised hierarchical clustering**. Unsupervised two-way hierarchical clustering was done with the software D-chip.

**Statistics**. Data analysis was performed using GraphPad Prism (version 8). Statistic tests and number of biological replicates (N) per each experiment are outlined in figure legends. Synergism was determined using the Synergyfinder webtool (https://synergyfinder.fimm.fi).

**Reporting summary**. Further information on research design is available in the Nature Research Reporting Summary linked to this article.

## Data availability
The exome seq, array CGH and DNA methylation data has been deposited in the database of genotypes and phenotypes (dbGaP) under the accession code phs002051.v1.p1. All numerical data underlying Figs. 1–6 and Supplementary Figs. 1–11 are provided as a source data file. All the other data supporting the findings of this study are available within the article and its supplementary information files and from the corresponding author upon reasonable request. A reporting summary for this article is available as a supplementary information file. PPCs can be obtained from the corresponding author through an MTA. All data associated with this study are present in the paper or the Supplementary Materials. Source data are provided with this paper.

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

## Acknowledgements

We acknowledge the expert technical help for animal experiments by Stephanie Kasper. We would especially like to thank M. Dyer and E. Steward for providing orthotopic PDX models from St Jude Children's Hospital through Childhood Solid Tumor Network. The work was supported by grants from the Swiss National Science Foundation (3100-156923 and 3100-175558), the Clinical Research Priority Program (CCRP) "Precision Heamatol-ogy/Oncology" and the Childhood Cancer Research Foundation Switzerland to BS. Concerning samples originating from PARIS, the PDX development was supported by the Société Française de Lutte contre les Cancers et les Leucémies de l'Enfant et l'Adolescent (Fondation Enfants et Santé), la Ligue Nationale Contre le Cancer, the Fondation AREMIG, and the Association Thibault BRIET, la Ligue Nationale Contre le Cancer and by the following grants: ERA-NET TRANSCAN JTC 2014 (TRAN201501238), TRANSCAN JTC 2017 (TRANS201801292) and H2020-lMI2-JTl-201 5-07 (116064 – ITCC P4). The MAPPYACTS protocol is supported by the Institut National du Cancer grant PHRC-K14–175, the Fondation ARC grant MAPY201501241 and Imagine For Margo.

## Author contributions

G.M., L.D.S., W.S., M.R., L.Z., D.S. and M.W. designed and performed experiments. G.M., M.W. and B.W.S. wrote the manuscript. Q.N. performed bioinformatical analyses. W.B.B., J.M., H.M., F.B., O.D., D.S. and B.R. provided primary RMS samples of PDX models. F.G.B., J.T., P.B., and F.K.N. supervised and designed experiments. M.W. and B.W.S. designed and supervised the study. All authors critically discussed the results and reviewed and approved the manuscript before submission.

## Competing interests

The authors declare no competing interests.
