## [Peer Review File · Nature Communications]

Reviewers' Comments:

Reviewer #1:

Remarks to the Author:

The manuscript by Manzella and colleagues presents a large series of experiments that characterize cells derived from cryopreserved rhabdomyosarcoma PDX's. Specifically, they show maintenance of genomic characteristics in short term culture, they optimize conditions for growth and show adverse effects of serum or growth factors. Using these 19 cultures they screen a library of FDA-approved drugs, and extend these findings to an in vivo model. Of interest is that 2D and 3D cultures give similar results, which is somewhat controversial, but perhaps correct.

It is proposed that this platform may be valuable for precision medicine. Importantly, a subset of RMS cultures are sensitive to MEK or Akt inhibition. It is claimed that Akt inhibition has not been a focus for treatment of RMS, although several inhibitors of Akt, TOR or RTKs that feed into PI3K/Akt have been evaluated clinically without significant antitumor activity being detected (either preclinical or clinical). It is of note, also, that the Dyer group at St.Jude have previously published use of cryopreserved PDX tissue as a screen for drug sensitivity. Unfortunately, several of the drugs, or combinations selected from in vitro testing did not show activity in PDX models. In most instances this is a result of being unable to attain adequate exposures to drug in vivo.

Thus, while the work presented is of interest, there is concern that the platform, like many others, essentially ignores much of the pharmacology of the drugs being tested. For example, protein binding is essentially ignored in the claims for clinically relevant exposures. For example, the heatmap presented in Figure 4A tested drugs at 500 nM. It is claimed that certain drugs had maximal toxicity against all RMS lines. This is hardly surprising, given that the concentration used is 20-30-fold above the IC50 concentration for some of these agents, when protein binding is at least partially represented.

In the one in vivo 'proof of concept' example given, afuresertib, the dose used in mice (100 mg/kg/day) probably far exceeds that exposures achieved in patients (dose 100 - 125 mg/day ~1.4 - 1.6 mg/kg). While C_{max} for afuresertib may be in the range (<281 nM), it is highly protein bound (probably >99% based on the long t_{1/2} in human), and B27 supplement to NB medium does not contain albumin as a potential drug-binding protein. Although afuresertib does inhibit growth of the early stage (<100mm³) IC-pPDX-35 xenograft, the tumor progresses on cycle 4. In a 'clinical' condition this would be progressive disease. For this platform to be considered as a functional approach to selecting effective therapeutics (i.e. those agents that cause tumor regressions), several additional examples, where later-stage tumors are shown to regress on the selected therapy would have to be presented. Further, drug exposures relevant to human exposure should be shown. As presented, the data in support of the platform, are not compelling.

Similarly, the comment that some RMS cells are exquisitely sensitive to MEK inhibition seems unfounded, when one compares the IC50 concentrations to trametinib, for example BRAF(V600E) glioma where the IC50's are in the 1-2 nM range in vitro. Even RAS-mutant FN-RMS really don't respond to MEK inhibitors due to feedback dis-inhibition of the MAPK pathway.

Finally, while screening programs that use a small number of tumors of any one type can be criticized (it would be more appropriate to cite the NCI 60 cell panel), the authors should consider approaches using single mouse designs that have been reported, and that are gaining acceptance in both pharma and academia. These designs allow incorporation of many PDX models to represent a disease state, hence more accurately encompass genetic and epigenetic heterogeneity.

In summary, there are interesting aspects to the study presented, but overall this manuscript does not provide compelling evidence that the platform is a major advance in the field.

Reviewer #2:

Remarks to the Author:

Review of Manzella et al., High throughput drug profiling with a living biobank of primary rhabdomyosarcoma cells unravels disease heterogeneity and detects an AKT inhibitor sensitive subgroup

In this study, the authors develop conditions for in vitro culture of primary rhabdomyosarcoma cells derived from PDX models (PPCs), and test the effects of small-molecule inhibitors on the cells. They investigate the effects of different culture conditions on viability, proliferation and differentiation of the cells, finding a striking effect of different media (for example, DMEM vs neurobasal) as well as growth factors such as bFGF. PPCs and the PDXs from which they derived were subjected to genomic profiling including exome sequencing, copy-number and methylation studies, generally revealing that the genetic makeup of the tumor cells was stable under the conditions used. After establishing the conditions for culture, they performed a high-throughput drug screen, which revealed heterogeneity in the response of different tumors to some of the agents. The most significant effects were seen with AKT and MEK inhibitors. In one case, they are able to establish a primary culture from a patient's malignant lymph node, testing 250 compounds for efficacy.

This is a well-designed study that has real translational significance for the treatment of RMS. The most important findings are the variable effects of different media, the lack of predictive power of genomic testing, and the potential for developing direct culture to support personalized medicine approaches. There are several weaknesses that should be addressed in order to increase the impact of the study. Throughout the manuscript, results are described in qualitative terms, and strong conclusions are drawn from these results. The arguments would be stronger if backed up with more rigorous statistics.

Specific comments:

1. How was the percentage of mouse cells quantified in Figure 1B? The methods do not give enough information on counting method, number of replicates scored, etc.
2. The finding of bFGF effect on cell proliferation is potentially significant. However, the link between bFGF, FGFR4 mutation/amplification status, and clinical response to inhibitors is not at all clear. The FGFR inhibitor used by the authors (CH5183284) inhibits FGFR1,2 and 3, but not FGFR4. The authors seem to imply (lines 413-419) that the inhibitory effect of bFGF in their assays suggests that FGFR inhibitors may paradoxically increase the proliferation of RMS cells, however this may not be the case. Understanding the mechanism of the bFGF effect is beyond the scope of the current study. However, in the absence of direct tests of this idea, the speculation should be qualified or softened.
3. Some observations seem tangential to the main thrust of the study. For example, the finding that spheroids are hypoxic in the center (Figure 1F), is expected and not particularly relevant here. Similarly, the addition of B-27 rescues proliferation of one PPC in the presence of FBS (lines 217-220), but this phenomenon is not explored further.
4. How many passages of the PPCs occurred before attempt were made to generate cell line-derived xenografts?
5. Figure S6: how was the histology evaluated and declared to be similar? How was the IHC scored? Please give statistics.
6. Figure 5 is an important result. Certainly, the ability to perform direct drug testing on primary

patient biopsy material could be clinically useful. There are a number of differences in the response; in general the primary LN-derived cells were more susceptible than the original PDX. Can the authors speculate on why, and what this means for their method? Were the conditions that were optimized for the PDX used for the primary tumor culture?

7. Figure 6B is confusing. What is the meaning of the triangles at right, are these opposing gradients in concentration of the two inhibitors? Please indicated the three groups that are supposedly shown here.

8. Figure 6, how were the # of floating cells quantified?

Reviewer #3:

Remarks to the Author:

This manuscript, by Manzella, Wachtel, and co-workers, describes the development of a screening platform derived from patient samples of rhabdomyosarcoma (RMS) tumors. While not a novel concept, as such screens are reported previously, this work is unusual in its complexity and completeness, as the authors went out of their way to investigate aspects of the platform in 2D, 3D, and PDX settings. Further, while the overall concept may not be new, this paper does extend the implementation of a patient-derived screening platform to a new tumor type, and the sensitivity to Akt inhibition may be translatable back into the clinic.

This paper much digs more deeply into optimized conditions for the culture of primary cancer cells compared to referenced studies. This is an extremely thorough study that may be enabling to future efforts in many tumor types. Characterization of the stromal tissue and identification of toxicity with several common conditions may serve as a warning to future researchers in related work. Better fidelity to the original sample (confirmed in this work using both histological and sequencing methods) is likely to lend more confidence in the reported results and, per the findings of the authors, to produce clones that better represent the initial tumor. An impactful finding of this study, for example, is that direct culture from patient samples is similar to use of an intermediary PDX – the requirement for in vivo implantation would introduce more time delays, which might limit or delay clinical benefit to identify precision medicines on a patient-by-patient basis.

In the abstract, the authors state that the responses to the screen were “surprisingly heterogeneous”. Given the stated diversity in RMS subtypes (esp ERMS), it is not clear why this would be surprising.

From the screen, the principal sensitivity followed-up on is that to Akt inhibitors, which was found in a select subgroup of samples. Notably, sensitivity to the AKT/PI3K axis was found in multiple similar studies referenced by the author (from references 7-11, at least 7, 8, and 10 mention sensitivity to mTOR/PI3K/Akt). Is this perhaps a general trend in the technique? Or an indication of widespread sensitivity to these inhibitors? Or just a coincidence? The follow-up on the Akt inhibition is generally well done, including exploration of combinations with MEK inhibition (also identified in the screen), and is complete with both efficacy and biomarker results. This would have been further enhanced by an in vivo combination with the MEKi.

In addition, this reviewer noted that at least one of the mTOR compounds (Torin 2) shows a roughly similar profile to the AKT inhibitors. It would be interesting to compare that compound with the Akt-targeted compounds for at least an in vitro follow-up.

Given that a range of compounds with various inhibition patterns on various targets were used, it is curious to me that the authors chose 500 nM of all inhibitors on the original screen. This may be well above the IC₅₀/IC₉₀ for some compounds on their primary biochemical target, but maybe only at or

near (or even below) the IC50/IC90 for others, especially given variances in cellular permeability often observed in molecules. Perhaps a better screening strategy may be to select a concentration for each inhibitor that correlates with either an achievable sustained dose in vivo or at least correlates to the potency of the molecule in a biochemical or cellular setting. Doing so may or may not yield different results, but might well wind up yielding even more relevant results.

Overall, this is a quite good manuscript, and is recommended for publication.

Typographical/grammatical feedback:

1. Abstract, first line. Would read better if phrased as "Cancer therapy is currently shifting from..."
2. Introduction, first line (79/80) would read better as "allows cures for many previously..."
3. Discussion, line 387, Perhaps "While" would be a better word choice than "Albeit"?

We would like to express our gratitude to the reviewers for their interest in our study as well as their very thorough and thoughtful comments. We have now addressed these raised concerns either by adopting the text or by providing novel experimental data. Most importantly, we provide novel *in vivo* data testing a combination of AKTi and MEKi that substantially strengthen our previous conclusions. More specifically, our point-by-point responses to the reviewer comments are detailed below:

Reviewers' comments:

Reviewer #1 (Expertise: RMS, therapy, Remarks to the Author):

The manuscript by Manzella and colleagues presents a large series of experiments that characterize cells derived from cryopreserved rhabdomyosarcoma PDX's. Specifically, they show maintenance of genomic characteristics in short term culture, they optimize conditions for growth and show adverse effects of serum or growth factors. Using these 19 cultures they screen a library of FDA-approved drugs, and extend these findings to an *in vivo* model. Of interest is that 2D and 3D cultures give similar results, which is somewhat controversial, but perhaps correct.

It is proposed that this platform may be valuable for precision medicine. Importantly, a subset of RMS cultures are sensitive to MEK or Akt inhibition. It is claimed that Akt inhibition has not been a focus for treatment of RMS, although several inhibitors of Akt, TOR or RTKs that feed into PI3K/Akt have been evaluated clinically without significant antitumor activity being detected (either preclinical or clinical). It is of note, also, that the Dyer group at St.Jude have previously published use of cryopreserved PDX tissue as a screen for drug sensitivity. Unfortunately, several of the drugs, or combinations selected from *in vitro* testing did not show activity in PDX models. In most instances this is a result of being unable to attain adequate exposures to drug *in vivo*.

-> The reviewer raises an important point here. The screen published by the Dyer group indeed used short term cultures from PDX with somewhat limited *in vivo* reproducibility. However, we believe that at least some of the discrepancies stem from use of non-optimized culture conditions. Our results indicate that addition of serum has a general negative effect on primary RMS cell growth which is exactly what the Dyer group has used. Hence, their drug effects might have been superimposed by a more general loss of viability due to culture conditions. We have now added new *in vivo* data using lower doses of AKTi combined with MEKi which clearly indicate that *in vitro* results are reproducible *in vivo* (see more details below).

Thus, while the work presented is of interest, there is concern that the platform, like many others, essentially ignores much of the pharmacology of the drugs being tested. For example, protein binding is essentially ignored in the claims for clinically relevant exposures. For example, the heatmap presented in Figure 4A tested drugs at 500 nM. It is claimed that certain drugs had maximal toxicity against all RMS lines. This is hardly surprising, given that the concentration used is 20-30-fold above the IC50 concentration for some of these agents, when protein binding is at least partially represented.

-> We agree that drug binding to (serum) proteins is an important issue. Importantly, for all the cultures of primary RMS cells, we use B-27 as serum replacement. B-27 actually contains albumin in similar amounts as standard 10% serum conditions, when used at 1x concentration (10% serum is 2.0-3.6 mg/ml, 1x B-27: 2.5 mg/ml (Podratz et al., Glia 1998). We used B-27 at 2x concentration as standard in NB medium and for the drug screening also in Adv.DMEM/F12 medium, to maintain comparable conditions. Hence, compared to standard medium containing 10% FBS, protein binding in our medium is represented certainly not inferior to standard conditions and well reflects drug exposures. We more specifically included this information in the description of our culture conditions in the Material and Methods section, and apologize if this point was unclear in our previous version:

"Media used for culture of PDX cells included DMEM (Sigma, D5671) or F10 (Life technologies, 11550-043) both supplemented with 10% heat inactivated FBS (Thermofisher Scientific), NB medium (Thermofisher Scientific, 21103049) supplemented with 2xB-27 (Thermofisher Scientific, 17504044), and Advanced DMEM/F12 (Thermofisher Scientific, 12634010) supplemented with 1x or 2x (for drug screening) B-27, 1.25 mM N-acetyl-L-cysteine (Sigma-Aldrich, A9165), 5 μ M A83-01 (Tocris Bioscience, 2939), and 10 μ M Y-27632 (Abmole Bioscience, M1817). All media were further supplemented with 100 U/ml penicillin/streptomycin and 2 mM glutamax (Life technologies, 35050-061)."

As noted by the reviewer, the 500 nM drug concentration used in our screening indeed might be high for some of the drugs, especially for targeted agents. However, an important aspect of our platform was establishment of culture conditions that allow continuous propagation of the cells. Hence, drug screening was used only for initial identification of interesting sensitivities which we then further validated with individual dose-response curves. Importantly, these conditions were clearly able to identify individual, patient-specific vulnerabilities for the majority of the drugs. Further, numerous multi-point drug-response curves are depicted in our manuscript, e.g. in Figure 4B/Supplemental Figure S8A-B (6 drugs and comparison of 2D and 3D cultures with 12 drugs), Supplemental Figure 6A (Idasanutlin), Supplemental Figure S8D (Torin-2), Figure 5E (combination of Panobinostat and Bortezomib), the new Figures 6C and D (MEK and AKT inhibitors) and the new Suppl. Figure 10A (combination of AKT and MEK inhibitors).

In the one *in vivo* 'proof of concept' example given, afuresertib, the dose used in mice (100 mg/kg/day) probably far exceeds that exposures achieved in patients (dose 100 - 125 mg/day \sim 1.4 - 1.6 mg/kg). While C_{max} for afuresertib may be in the range (<281 nM), it is highly protein bound (probably $>99\%$ based on the long $t_{1/2}$ in human), and B27 supplement to NB medium does not contain albumin as a potential drug-binding protein. Although afuresertib does inhibit growth of the early stage (<100 mm³) IC-pPDX-35 xenograft, the tumor progresses on cycle 4. In a 'clinical' condition this would be progressive disease. For this platform to be considered as a functional approach to selecting effective therapeutics (i.e. those agents that cause tumor regressions), several additional examples, where later-stage tumors are shown to regress on the selected therapy would have to be presented. Further, drug exposures relevant to human exposure should be shown. As presented, the data in support of the platform, are not compelling.

-> As commented above, the B27 supplement indeed does contain albumin.

Albeit one important aspect of this experiment was to test whether the differences in response found *in vitro* reflect the response *in vivo* (which was indeed the case), we agree with the reviewer that the 100 mg/kg/day dose of afuresertib is above the MTD for humans determined in previously published clinical phase I studies. Hence, we repeated the *in vivo* experiment with lower doses of afuresertib fulfilling this criteria in combination with trametinib. Based on the 125 mg/day MTD for humans, we calculated the corresponding concentration for mice (125 mg/day for a 70 kg person equals 22 mg/kg per day for a mouse) applying a commonly used formula for dose conversions between humans and mice (Nair AB and Jacob S, J Basic Clin Pharm 2016).

Similarly, the comment that some RMS cells are exquisitely sensitive to MEK inhibition seems unfounded, when one compares the IC_{50} concentrations to trametinib, for example BRAF(V600E) glioma where the IC_{50} 's are in the 1-2 nM range *in vitro*. Even RAS-mutant FN-RMS really don't respond to MEK inhibitors due to feedback dis-inhibition of the MAPK pathway.

-> We attempted to describe differences in sensitivity among the different RMS models and not to other cancer types in general. Since this might be confusing, we replaced the term "exquisite" by "selective".

Nevertheless, in a recent publication (Yohe et al. Science Transl Med 2018) the MEK inhibitor trametinib was shown to control tumor growth *in vivo* in two eRMS xenograft models (slight

regression or stable disease over some time, before resistance developed), suggesting that for some eRMS there might be a therapeutic window.

Therefore, we now performed dose-response analysis with trametinib in 8 of our primary cultures. This data is now shown in the new Figure 6C. It shows that the IC50 in the most sensitive case (N870) is 4 nM, about the same range reported for glioma and a second case also responded with a low IC50 (7 nM). Interestingly, while these cases are RAS mutant cases, some other RAS mutant cases responded differently with IC50s of 32 and 238 nM. In these cases, many of the cells differentiated (cytostatic and not cytotoxic response), which is depicted in the new Suppl. Figures 10B and C. This detailed analysis again revealed patient-specific heterogeneity and highlights the relevance of our drug profiling approach.

Finally, while screening programs that use a small number of tumors of any one type can be criticized (it would be more appropriate to cite the NCI 60 cell panel), the authors should consider approaches using single mouse designs that have been reported, and that are gaining acceptance in both pharma and academia. These designs allow incorporation of many PDX models to represent a disease state, hence more accurately encompass genetic and epigenetic heterogeneity.

-> We agree with the reviewer that single mouse approaches are indeed an alternative. However, we believe that drug testing in a co-clinical setting directly *in vivo* will be challenging due to time constraints (some of our PDX needed several month until initial growth could be observed) and the number of drugs that can be tested is clearly limited by logistics. Hence, the approach describe here should be seen as complementary, with clear advantages when primary cultures can be directly established from tumor resections. It was never our intention to criticize the concept of drug testing programs using small numbers of mice. We adjusted the wording in the discussion which now reads: "Heterogeneity at different levels strongly argues for a functional precision medicine approach and might be taken into account in screening programs using a fixed number of models (38). Importantly, experimental-designs using one mouse per tumor have been proposed as approach to test a large number of different tumors *in vivo* in parallel (Ref Ghilu et al., Cancer Chemotherapy and Pharmacology 2020) and might represent an alternative.

In summary, there are interesting aspects to the study presented, but overall this manuscript does not provide compelling evidence that the platform is a major advance in the field.

Reviewer #2 (Expertise: RMS models and genomics, Remarks to the Author):

Review of Manzella et al., High throughput drug profiling with a living biobank of primary rhabdomyosarcoma cells unravels disease heterogeneity and detects an AKT inhibitor sensitive subgroup

In this study, the authors develop conditions for *in vitro* culture of primary rhabdomyosarcoma cells derived from PDX models (PPCs), and test the effects of small-molecule inhibitors on the cells. They investigate the effects of different culture conditions on viability, proliferation and differentiation of the cells, finding a striking effect of different media (for example, DMEM vs neurobasal) as well as growth factors such as bFGF. PPCs and the PDXs from which they derived were subjected to genomic profiling including exome sequencing, copy-number and methylation studies, generally revealing that the genetic makeup of the tumor cells was stable under the conditions used. After establishing the conditions for culture, they performed a high-throughput drug screen, which revealed heterogeneity in the response of different tumors to some of the agents. The most significant effects were seen with AKT and MEK inhibitors. In one case, they are able to establish a primary culture from a patient's malignant lymph node, testing 250 compounds for efficacy.

This is a well-designed study that has real translational significance for the treatment of RMS. The most important findings are the variable effects of different media, the lack of predictive power of genomic testing, and the potential for developing direct culture to support personalized medicine approaches.

We would like to thank the reviewer for these supportive comments.

There are several weaknesses that should be addressed in order to increase the impact of the study. Throughout the manuscript, results are described in qualitative terms, and strong conclusions are drawn from these results. The arguments would be stronger if backed up with more rigorous statistics.

Specific comments:

1. How was the percentage of mouse cells quantified in Figure 1B? The methods do not give enough information on counting method, number of replicates scored, etc.

-> For quantification of mouse cells, we used DAPI stained cultures from at least two (N=2-3) independent (=originating from two different tumors) PDX cultures. DAPI stained mouse cells are readily identified by heterochromatic dots as described in supplemental Figure S1C. Visual fields with at least 200 cells were selected and mouse and human cells counted manually. We now included the following paragraph in the Material and Methods part of the manuscript describing this procedure:

"Quantification of mouse cells

For quantification of mouse stroma cells present in PPC cultures, cells isolated from PDX and grown under different conditions in 96 well plates were stained with DAPI. Mouse cells were then identified on microscopic images by the punctate DAPI-staining pattern of their nuclei. Relative numbers of human and mouse cells were determined on microscopic images containing at least 200 cells if possible."

2. The finding of bFGF effect on cell proliferation is potentially significant. However, the link between bFGF, FGFR4 mutation/amplification status, and clinical response to inhibitors is not at all clear. The FGFR inhibitor used by the authors (CH5183284) inhibits FGFR1,2 and 3, but not FGFR4. The authors seem to imply (lines 413-419) that the inhibitory effect of bFGF in their assays suggests that FGFR inhibitors may paradoxically increase the proliferation of RMS cells, however this may not be the case. Understanding the mechanism of the bFGF effect is beyond the scope of the current study. However, in the absence of direct tests of this idea, the speculation should be qualified or softened.

-> We completely agree with all these insightful thoughts of the reviewer. Therefore, we changed the text to soften our conclusion. We now write "...suggesting that the signaling downstream of the FGFRs in RMS cells is more complex than previously appreciated and might have anti-proliferative effects under some circumstances."

3. Some observations seem tangential to the main thrust of the study. For example, the finding that spheroids are hypoxic in the center (Figure 1F), is expected and not particularly relevant here. Similarly, the addition of B-27 rescues proliferation of one PPC in the presence of FBS (lines 217-220), but this phenomenon is not explored further.

-> We agree with the reviewer that morphology of the spheroids including the hypoxic core is as expected. However, we decided to include this data to emphasize that our 3D cultures are performing as expected and have characteristics as described elsewhere. This was done in light of our findings that there were no significant differences in drug response between 2D and 3D (spheroid) cultures, a result that might be less expected. However, we completely agree that the description of the spheroids is not a surprising result and moved this data to a new supplementary Figure 3.

Regarding the protective effect of B-27 in one PPC cultured in presence of serum, we also agree with the reviewer that this phenomenon is less relevant for the study. We therefore removed the data (lowest panel of Figure 2D and corresponding text on page 6) from the manuscript.

4. How many passages of the PPCs occurred before attempt were made to generate cell line-derived xenografts?

-> We used passages 4-6 of the cultured cells to compare the engraftment with dissociated parental PDX. We included this information in the text on page 7 and in the legend of Figure 1F.

5. Figure S6: how was the histology evaluated and declared to be similar? How was the IHC scored? Please give statistics.

-> Histology was qualitatively evaluated by a pathologist (Peter Bode). H&E stainings revealed round cell morphologies for most aRMS, while a more spindle-cell morphology was detected in case of eRMS, which is in line with histological features characteristic for human RMS tumors. However, since the number of available sections from primary tumors was small (samples were collected from different European institutions complicating access to paraffin material), we could only in two cases directly compare the PDX with human tumors (we added the one for RMS-ZH003 in Suppl. Figure 7A).

IHC was indeed not scored in the originally submitted manuscript, therefore we now have done this and include scoring of the IHC pictures as percentage of positive cells. This data is depicted in the new Suppl. Figure 7B and shows that Myogenin positivity discriminates ARMS from ERMS with high significance, with most ARMS containing more than 50% positive cells, while most ERMS have less than 50 percent positive cells. These numbers are in good agreement with available literature, where Myogenin has been proposed as marker to discriminate ARMS from ERMS tumors (Dias P et al., Am J Pathol. 2000). In contrast, most PDX are composed of nearly 100 percent Desmin positive cells.

Based on the limitations mentioned above, we changed the conclusion of the histological analysis on page 7-8 to:

"Impressively, both PDX and CDX show characteristic RMS architecture and a degree of MYOGENIN and DESMIN positivity, which is in line with published data showing that number of MYOGENIN positive cells discriminates ARMS from ERMS (Suppl. Figure S6A and S6B)."

6. Figure 5 is an important result. Certainly, the ability to perform direct drug testing on primary patient biopsy material could be clinically useful. There are a number of differences in the response; in general the primary LN-derived cells were more susceptible than the original PDX. Can the authors speculate on why, and what this means for their method? Were the conditions that were optimized for the PDX used for the primary tumor culture?

-> Differences in response between PDX and patient-derived cultures could be generated by different aspects. First, the PDX-derived cells originated from the first relapse, whereas the direct culture was established from the second relapse (lymph node metastasis). Furthermore, these specific cells were quite sensitive towards passaging and tended to differentiate very easily which was probably an important aspect influencing the drug response.

The conditions that were used for the primary tumor-derived culture and the PDX-derived culture were slightly different, with the NB medium mixture used for the patient cells and the Adv.DMEM/F12 mixture for the PDX cells.

7. Figure 6B is confusing. What is the meaning of the triangles at right, are these opposing gradients in concentration of the two inhibitors? Please indicated the three groups that are supposedly shown here.

-> The triangles on the right were supposed to illustrate the sensitivity towards the MEK inhibitor (red) and the AKT inhibitor (blue). We realized that the text "MEK inhibition" and "AKT inhibition" in the triangles is misleading as well as was the orientation of the triangles. We therefore changed the text to "MEKi sensitivity" and "AKTi sensitivity" and updated the Figure accordingly.

8. Figure 6, how were the # of floating cells quantified?

-> These pictures should illustrate qualitatively the morphological change upon treatment with Afuresertib. Hence, originally we did not further quantify the number of floating cells. We now repeated the experiment and included cell counts in the Figure, which is now in the supplemental part as Suppl. Figure 11B.

Reviewer #3 (Expertise: Drug screening, cancer, Remarks to the Author):

This manuscript, by Manzella, Wachtel, and co-workers, describes the development of a screening platform derived from patient samples of rhabdomyosarcoma (RMS) tumors. While not a novel concept, as such screens are reported previously, this work is unusual in its complexity and completeness, as the authors went out of their way to investigate aspects of the platform in 2D, 3D, and PDX settings. Further, while the overall concept may not be new, this paper does extend the implementation of a patient-derived screening platform to a new tumor type, and the sensitivity to Akt inhibition may be translatable back into the clinic.

This paper much digs more deeply into optimized conditions for the culture of primary cancer cells compared to referenced studies. This is an extremely thorough study that may be enabling to future efforts in many tumor types. Characterization of the stromal tissue and identification of toxicity with several common conditions may serve as a warning to future researchers in related work. Better fidelity to the original sample (confirmed in this work using both histological and sequencing methods) is likely to lend more confidence in the reported results and, per the findings of the authors, to produce clones that better represent the initial tumor. An impactful finding of this study, for example, is that direct culture from patient samples is similar to use of an intermediary PDX – the requirement for *in vivo* implantation would introduce more time delays, which might limit or delay clinical benefit to identify precision medicines on a patient-by-patient basis.

We would like to thank the reviewer for his supportive comments.

In the abstract, the authors state that the responses to the screen were “surprisingly heterogeneous”. Given the stated diversity in RMS subtypes (esp ERMS), it is not clear why this would be surprising.

-> We removed the term “surprisingly”.

From the screen, the principal sensitivity followed-up on is that to Akt inhibitors, which was found in a select subgroup of samples. Notably, sensitivity to the AKT/PI3K axis was found in multiple similar studies referenced by the author (from references 7-11, at least 7, 8, and 10 mention sensitivity to mTor/PI3K/Akt). Is this perhaps a general trend in the technique? Or an indication of widespread sensitivity to these inhibitors? Or just a coincidence? The follow-up on the Akt inhibition is generally well done, including exploration of combinations with MEK inhibition (also identified in the screen), and is complete with both efficacy and biomarker results. This would have been further enhanced by an *in vivo* combination with the MEKi.

-> We agree that an *in vivo* study of the combination of AKT and MEK inhibitors would complement our dataset. Therefore, we now performed such an additional experiment using a combination of afuresertib and trametinib. As suggested by reviewer #1, we used the two drugs in concentrations applicable to human patients, hence afuresertib was used in a 5x lower concentration than in the *in vivo* experiment shown in Fig 6F. This novel data is depicted in Fig.6G and shows that the two drugs indeed display synergy.

As suggested by the editor, we also tested this drug combination *in vitro* with 8 different PPCs and detected in most cases a synergistic behavior of the two drugs (as determined by Bliss synergy analysis). Especially in case of some RAS mutant eRMS, very high synergy scores (Bliss scores 10-

20) were detected. However, among the *RAS* mutant eRMS (4 cases) there is still a significant heterogeneity of responses with IC50 values of Trametinib of 4, 7.5, 32, and 238 nM. This heterogeneity is at least partially explained by the ratio of cells responding with cytotoxicity vs cytostatics (differentiation), which decreases with increasing IC50. To evaluate this further, we determined the number of Myosin Heavy Chain positive cells in the most resistant case by high throughput microscopy. This data is depicted in the new suppl. Figures 10B and C. All together, these findings again highlight the relevance of our profiling strategy for drug selection.

In addition, this reviewer noted that at least one of the mTOR compounds (Torin 2) shows a roughly similar profile to the AKT inhibitors. It would be interesting to compare that compound with the Akt-targeted compounds for at least an *in vitro* follow-up.

-> Thank you very much for this observation. However, differences in response towards torin-2 were much smaller than in case of AKT inhibitors (remaining cell viabilities were for torin-2: 10/21 samples <30% and 20/21 samples <60%; and for afuresertib: 5/21 samples <30% and 10/21 samples <60%). Nevertheless, we still considered the suggested experiment to be of interest. Hence, we performed a detailed dose-response study of torin-2 with 7 different PPC cultures *in vitro*. The data is shown in the new suppl. Fig. 9D and demonstrates that, in contrast to MEK and AKT inhibitors, response to torin-2 is much more uniform for these 7 samples, with IC50 values all lying in a concentration range between 13 and 70 nM.

Given that a range of compounds with various inhibition patterns on various targets were used, it is curious to me that the authors chose 500 nM of all inhibitors on the original screen. This may be well above the IC50/IC90 for some compounds on their primary biochemical target, but maybe only at or near (or even below) the IC50/IC90 for others, especially given variances in cellular permeability often observed in molecules. Perhaps a better screening strategy may be to select a concentration for each inhibitor that correlates with either an achievable sustained dose *in vivo* or at least correlates to the potency of the molecule in a biochemical or cellular setting. Doing so may or may not yield different results, but might well wind up yielding even more relevant results.

->Our screening procedure was originally designed for identification of interesting sensitivities in individual cases compared to the entire set of samples. Since our platform includes establishment of culture conditions that allow continuous propagation of the cells, the idea was to select interesting drugs and perform a full dose response curve in a second step (as depicted in Supplemental Figure S9A and S9D (dose response curves of 6 drugs and Torin-2), Figure 4B/Supplemental Figure S9B (IC50 values and dose response curves for 12 drugs determined with 2D and 3D cultures) and Figures 6C and 6D (dose response curves of one MEK and two AKT inhibitors).

Overall, this is a quite good manuscript, and is recommended for publication.

Typographical/grammatical feedback:

1. Abstract, first line. Would read better if phrased as "Cancer therapy is currently shifting from..."
2. Introduction, first line (79/80) would read better as "allows cures for many previously..."
3. Discussion, line 387, Perhaps "While" would be a better word choice than "Albeit"?

-> Thank you very much for these suggestions. We changed all these sentences accordingly.

Reviewers' Comments:

Reviewer #1:

Remarks to the Author:

The authors have responded to my previous concerns, but unfortunately have not really addressed the critical issues raised.

In this revision, the authors have included additional information particularly concerning the in vitro to in vivo activity of combination treatments. While the data are of interest, the activity of afisertib +trametinib is still progressive disease (using clinical criteria). Several groups have shown the activity of combined MAPK inhibitors having activity against RAS-mutant embryonal rhabdomyosarcomas (some lines are indeed very responsive), so this aspect is not particularly novel.

However, I feel that although a great deal of valuable information is presented (e.g. growth conditions for propagating cell lines), this work does not represent a conceptual leap forward. The authors present 21 RMS cell lines. Why 21? Are 11 fusion-negative lines adequate to represent the heterogeneity/diversity and similarly 10 fusion-positive lines - same question.

The data presented in Fig.4A is also troubling. HSP90 and proteasome inhibitors have demonstrated no activity against rhabdomyosarcoma models (or clinically), whereas bortezomid is quite active against ALL PDX models. Thus, it is unclear how the screen is of value - for example it does not separate bortezomid from the other proteasome inhibitors. Similarly, Torin 2 would be an agent selected from the screen yet TOR kinase inhibitors appear to have less in vivo activity than do rapalogs (everolimus is inactive in the screen presented).

For the proposed in vitro screen to be valuable, the authors would need to demonstrate that based on the screen they could identify multiple agents that induced tumor regressions when tested against the respective PDX.

Rhabdomyosarcoma is treated with essentially 6 DNA damaging agents + vincristine (V): cyclophosphamide/actinomycin D (VAC) irinotecan (I), doxorubicin, etoposide and ifosfamide + radiation. Both VAC and VI give similar response rates (65-72% objective responses), yet these agents, when tested, have quite limited activity in the in vitro screen. Perhaps testing of the known actives (activated cyclophosphamide or SN-38) would give more confidence in the predictions of this assay?

Reviewer #2:

Remarks to the Author:

The revised manuscript "High throughput drug profiling with a living biobank of primary rhabdomyosarcoma cells unravels disease heterogeneity and detects an AKT inhibitor sensitive subgroup" by Manzella et al is a substantial improvement on the original version. The new data in Figure 6 are particularly valuable. At this point, the authors have satisfied my concerns.

There are a couple of small typographical errors in Figure 5

Figure 5A: suggest "pulmonary metastases" rather than "pumonal". Also, drug "profile" rather than "profil"

Reviewer #3:

Remarks to the Author:

Manzella, Wachtel, and co-workers present a revised manuscript that includes additional data, including, importantly, new in vivo results. I feel that the authors have adequately addressed the concerns of the reviewers and recommend that this manuscript be published.

Additional detail on two significant topics from the initial review:

- Two of the reviewers stated concerns regarding the "one-size-fits-all" 500 nM doses of the compounds in the screen, without specific regard to compound differences with respect to other pharmacology, protein binding, permeability, etc. [It is noted that this is a difficult task and consideration in many screening exercises.] While an improvement in the process can likely be realized by future consideration of such factors and perhaps better tailoring doses of each specific compound, even in the original screen, the concerns of the reviewers are likely adequately addressed.
- The new data presented in Figure 6 with the more relevant and lower doses of the combination are powerful and certainly strengthen the paper.

Suggestions for improvement/corrections:

Line 375: "proofed" likely should be "proved"

Line 439: "affect" should be "effect"

Figure 6G: I suggest to change "low dose" to the actual doses used of the compounds

REVISION 1:

Whilst we would expect you to address all reviewer concerns in full, in particular, we suggest a) comparing the potency for some of your hits in an albumin containing media, b) test lower doses of Akt inhibitor in vitro and in vivo in established tumours, in combination with MEKi, c) address all the other technical concerns of ref#2 and #3.

We would like to express our gratitude to the reviewers and the associate editor of Nature Communications for their interest in our study as well as their very thorough and thoughtful comments. We have now addressed all previously raised concerns either by adopting the text or by providing novel experimental data. Most importantly, as suggested by the editor, we provide novel *in vivo* data testing a combination of AKTi and MEKi that substantially strengthen our previous conclusions. More specifically, our point-by-point responses to the reviewer comments are detailed below:

Reviewers' comments:

Reviewer #1 (Expertise: RMS, therapy, Remarks to the Author):

The manuscript by Manzella and colleagues presents a large series of experiments that characterize cells derived from cryopreserved rhabdomyosarcoma PDX's. Specifically, they show maintenance of genomic characteristics in short term culture, they optimize conditions for growth and show adverse effects of serum or growth factors. Using these 19 cultures they screen a library of FDA-approved drugs, and extend these findings to an in vivo model. Of interest is that 2D and 3D cultures give similar results, which is somewhat controversial, but perhaps correct.

It is proposed that this platform may be valuable for precision medicine. Importantly, a subset of RMS cultures are sensitive to MEK or Akt inhibition. It is claimed that Akt inhibition has not been a focus for treatment of RMS, although several inhibitors of Akt, TOR or RTKs that feed into PI3K/Akt have been evaluated clinically without significant antitumor activity being detected (either preclinical or clinical). It is of note, also, that the Dyer group at St.Jude have previously published use of cryopreserved PDX tissue as a screen for drug sensitivity. Unfortunately, several of the drugs, or combinations selected from in vitro testing did not show activity in PDX models. In most instances this is a result of being unable to attain adequate exposures to drug in vivo.

-> The reviewer raises an important point here. The screen published by the Dyer group indeed used short term cultures from PDX with somewhat limited *in vivo* reproducibility. However, we believe that at least some of the discrepancies stem from use of non-optimized culture conditions. Our results indicate that addition of serum has a general negative effect on primary RMS cell growth which is exactly what the Dyer group has used. Hence, their drug effects might have been superimposed by a more general loss of viability due to culture conditions. We have now added new *in vivo* data using lower doses of AKTi combined with MEKi which clearly indicate that *in vitro* results are reproducible *in vivo* (see more details below).

Thus, while the work presented is of interest, there is concern that the platform, like many others, essentially ignores much of the pharmacology of the drugs being tested. For example, protein binding is essentially ignored in the claims for clinically relevant exposures. For example, the heatmap presented in Figure 4A tested drugs at 500 nM. It is claimed that certain drugs had maximal toxicity against all RMS lines. This is hardly surprising, given that the concentration used is 20-30-fold above the IC50 concentration for some of these agents, when protein binding is at least partially represented.

-> We agree that drug binding to (serum) proteins is an important issue. Importantly, for all the cultures of primary RMS cells, we use B-27 as serum replacement. B-27 actually contains albumin in similar amounts as standard 10% serum conditions, when used at 1x concentration (10% serum

is 2.0-3.6 mg/ml, 1x B-27: 2.5 mg/ml (Podratz et al., Glia 1998). We used B-27 at 2x concentration as standard in NB medium and for the drug screening also in Adv.DMEM/F12 medium, to maintain comparable conditions. Hence, compared to standard medium containing 10% FBS, protein binding in our medium is represented certainly not inferior to standard conditions and well reflects drug exposures. We more specifically included this information in the description of our culture conditions in the Material and Methods section, and apologize if this point was unclear in our previous version:

"Media used for culture of PDX cells included DMEM (Sigma, D5671) or F10 (Life technologies, 11550-043) both supplemented with 10% heat inactivated FBS (Thermofisher Scientific), NB medium (Thermofisher Scientific, 21103049) supplemented with 2xB-27 (Thermofisher Scientific, 17504044), and Advanced DMEM/F12 (Thermofisher Scientific, 12634010) supplemented with 1x or 2x (for drug screening) B-27, 1.25 mM N-acetyl-L-cysteine (Sigma-Aldrich, A9165), 5 μ M A83-01 (Tocris Bioscience, 2939), and 10 μ M Y-27632 (Abmole Bioscience, M1817). All media were further supplemented with 100 U/ml penicillin/streptomycin and 2 mM glutamax (Life technologies, 35050-061)."

As noted by the reviewer, the 500 nM drug concentration used in our screening indeed might be high for some of the drugs, especially for targeted agents. However, an important aspect of our platform was establishment of cultures conditions that allow continuous propagation of the cells. Hence, drug screening was used only for initial identification of interesting sensitivities which we then further validated with individual dose-response curves. Importantly, these conditions were clearly able to identify individual, patient-specific vulnerabilities for the majority of the drugs. Further, numerous multi-point drug-response curves are depicted in our manuscript, e.g. in Figure 4B/Supplemental Figure S9, Figure 5E (combination treatment), the new Figures 6C and D (MEK and AKT inhibitors) and the new Suppl. Figure 10A (combination of AKT and MEK inhibitors).

In the one *in vivo* 'proof of concept' example given, afuresertib, the dose used in mice (100 mg/kg/day) probably far exceeds that exposures achieved in patients (dose 100 - 125 mg/day ~1.4 - 1.6 mg/kg). While C_{max} for afuresertib may be in the range (<281 nM), it is highly protein bound (probably >99% based on the long t_{1/2} in human), and B27 supplement to NB medium does not contain albumin as a potential drug-binding protein. Although afuresertib does inhibit growth of the early stage (<100mm³) IC-pPDX-35 xenograft, the tumor progresses on cycle 4. In a 'clinical' condition this would be progressive disease. For this platform to be considered as a functional approach to selecting effective therapeutics (i.e. those agents that cause tumor regressions), several additional examples, where later-stage tumors are shown to regress on the selected therapy would have to be presented. Further, drug exposures relevant to human exposure should be shown. As presented, the data in support of the platform, are not compelling.

-> As commented above, the B27 supplement indeed does contain albumin.

Albeit one important aspect of this experiment was to test whether the differences in response found *in vitro* reflect the response *in vivo* (which was indeed the case), we agree with the reviewer that the 100 mg/kg/day dose of afuresertib is above the MTD for humans determined in previously published clinical phase I studies. Hence, we repeated the *in vivo* experiment with lower doses of afuresertib fulfilling this criteria in combination with trametinib. Based on the 125 mg/day MTD for humans, we calculated the corresponding concentration for mice (125 mg/day for a 70 kg person equals 22 mg/kg per day for a mouse) applying a commonly used formula for dose conversions between humans and mice (Nair AB and Jacob S, J Basic Clin Pharm 2016).

Similarly, the comment that some RMS cells are exquisitely sensitive to MEK inhibition seems unfounded, when one compares the IC₅₀ concentrations to trametinib, for example BRAF(V600E) glioma where the IC₅₀'s are in the 1-2 nM range *in vitro*. Even RAS-mutant FN-RMS really don't respond to MEK inhibitors due to feedback dis-inhibition of the MAPK pathway.

-> We attempted to describe differences in sensitivity among the different RMS models and not to other cancer types in general. Since this might be confusing, we replaced the term "exquisite" by "selective".

Nevertheless, in a recent publication (Yohe et al. Science Transl Med 2018) the MEK inhibitor trametinib was shown to control tumor growth *in vivo* in two eRMS xenograft models (slight regression or stable disease over some time, before resistance developed), suggesting that for some eRMS there might be a therapeutic window.

Therefore, we now performed dose-response analysis with trametinib with 8 of our primary cultures. This data is now shown in the new Figure 6C. It shows that the IC50 in the most sensitive case (N870) is 4 nM, about the same range reported for glioma and a second case also responded with a low IC50 (7 nM). Interestingly, while these cases are RAS mutant cases, some other RAS mutant cases responded differently with IC50s of 32 and 238 nM. In these cases, many of the cells differentiated (cytostatic and not cytotoxic response), which is depicted in the new Suppl. Figures 10B and C. This detailed analysis again revealed patient-specific heterogeneity and highlights the relevance of our drug profiling approach.

Finally, while screening programs that use a small number of tumors of any one type can be criticized (it would be more appropriate to cite the NCI 60 cell panel), the authors should consider approaches using single mouse designs that have been reported, and that are gaining acceptance in both pharma and academia. These designs allow incorporation of many PDX models to represent a disease state, hence more accurately encompass genetic and epigenetic heterogeneity.

-> We agree with the reviewer that single mouse approaches are indeed an alternative. However, we believe that drug testing in a co-clinical setting directly *in vivo* will be challenging due to time constraints (some of our PDX needed several month until initial growth could be observed) and the number of drugs that can be tested is clearly limited by logistics. Hence, the approach describe here should be seen as complementary, with clear advantages when primary cultures can be directly established from tumor resections. It was never our intention to criticize the concept of drug testing programs using small numbers of mice. We adjusted the wording in the discussion which now reads: "Heterogeneity at different levels strongly argues for a functional precision medicine approach and might be taken into account in screening programs using a fixed number of models (38). Importantly, experimental-designs using one mouse per tumor have been proposed as approach to test a large number of different tumors *in vivo* in parallel (Ref Ghilu et al., Cancer Chemotherapy and Pharmacology 2020) and might represent an alternative.

In summary, there are interesting aspects to the study presented, but overall this manuscript does not provide compelling evidence that the platform is a major advance in the field.

Reviewer #2 (Expertise: RMS models and genomics, Remarks to the Author):

Review of Manzella et al., High throughput drug profiling with a living biobank of primary rhabdomyosarcoma cells unravels disease heterogeneity and detects an AKT inhibitor sensitive subgroup

In this study, the authors develop conditions for *in vitro* culture of primary rhabdomyosarcoma cells derived from PDX models (PPCs), and test the effects of small-molecule inhibitors on the cells. They investigate the effects of different culture conditions on viability, proliferation and differentiation of the cells, finding a striking effect of different media (for example, DMEM vs neurobasal) as well as growth factors such as bFGF. PPCs and the PDXs from which they derived were subjected to genomic profiling including exome sequencing, copy-number and methylation studies, generally revealing that the genetic makeup of the tumor cells was stable under the

conditions used. After establishing the conditions for culture, they performed a high-throughput drug screen, which revealed heterogeneity in the response of different tumors to some of the agents. The most significant effects were seen with AKT and MEK inhibitors. In one case, they are able to establish a primary culture from a patient's malignant lymph node, testing 250 compounds for efficacy.

This is a well-designed study that has real translational significance for the treatment of RMS. The most important findings are the variable effects of different media, the lack of predictive power of genomic testing, and the potential for developing direct culture to support personalized medicine approaches.

We would like to thank the reviewer for these supportive comments.

There are several weaknesses that should be addressed in order to increase the impact of the study. Throughout the manuscript, results are described in qualitative terms, and strong conclusions are drawn from these results. The arguments would be stronger if backed up with more rigorous statistics.

Specific comments:

1. How was the percentage of mouse cells quantified in Figure 1B? The methods do not give enough information on counting method, number of replicates scored, etc.

-> For quantification of mouse cells, we used DAPI stained cultures from at least two (N=2-3) independent (=originating from two different tumors) PDX cultures. DAPI stained mouse cells are readily identified by heterochromatic dots as described in supplemental Figure S1C. Visual fields with at least 200 cells were selected and mouse and human cells counted manually. We now included the following paragraph in the Material and Methods part of the manuscript describing this procedure:

" Quantification of mouse cells

For quantification of mouse stroma cells present in PPC cultures, cells isolated from PDX and grown under different conditions in 96 well plates were stained with DAPI. Mouse cells were then identified on microscopic images by the punctate DAPI-staining pattern of their nuclei. Relative numbers of human and mouse cells were determined on microscopic images containing at least 200 cells if possible."

2. The finding of bFGF effect on cell proliferation is potentially significant. However, the link between bFGF, FGFR4 mutation/amplification status, and clinical response to inhibitors is not at all clear. The FGFR inhibitor used by the authors (CH5183284) inhibits FGFR1,2 and 3, but not FGFR4. The authors seem to imply (lines 413-419) that the inhibitory effect of bFGF in their assays suggests that FGFR inhibitors may paradoxically increase the proliferation of RMS cells, however this may not be the case. Understanding the mechanism of the bFGF effect is beyond the scope of the current study. However, in the absence of direct tests of this idea, the speculation should be qualified or softened.

-> We completely agree with all these insightful thoughts of the reviewer. Therefore, we changed the text to soften our conclusion. We now write "...suggesting that the signaling downstream of the FGFRs in RMS cells is more complex than previously appreciated and might have anti-proliferative effects under some circumstances."

3. Some observations seem tangential to the main thrust of the study. For example, the finding that spheroids are hypoxic in the center (Figure 1F), is expected and not particularly relevant here. Similarly, the addition of B-27 rescues proliferation of one PPC in the presence of FBS (lines 217-220), but this phenomenon is not explored further.

-> We agree with the reviewer that morphology of the spheroids including the hypoxic core is as expected. However, we decided to include this data to emphasize that our 3D cultures are

performing as expected and have characteristics as described elsewhere. This was done in light of our findings that there were no significant differences in drug response between 2D and 3D (spheroid) cultures, a result that might be less expected. However, we completely agree that the description of the spheroids is not a surprising result and moved this data to a new supplementary Figure 3.

Regarding the protective effect of B-27 on one PPC cultured in presence of serum, we also agree with the reviewer that this phenomenon is less relevant for the study. We therefore removed the data (lowest panel of Figure 2D and corresponding text on page 6) from the manuscript.

4. How many passages of the PPCs occurred before attempt were made to generate cell line-derived xenografts?

-> We used passages 4-6 of the cultured cells to compare the engraftment with dissociated parental PDX. We included this information in the text on page 7 and in the legend of Figure 1F.

5. Figure S6: how was the histology evaluated and declared to be similar? How was the IHC scored? Please give statistics.

-> Histology was qualitatively evaluated by a pathologist (Peter Bode). H&E stainings revealed round cell morphologies for most aRMS, while a more spindle-cell morphology was detected in case of eRMS, which is in line with histological features characteristic for human RMS tumors. However, since the number of available sections from primary tumors was small (samples were collected from different European institutions complicating access to paraffin material), we could only in two cases directly compare the PDX with human tumors (we added the one for RMS-ZH003 in Suppl. Figure 7A).

As IHC was indeed not scored in the originally submitted manuscript, but we now have done this and include scoring of the IHC pictures as percentage of positive cells. This data is depicted in the new Suppl. Figure 7B and shows that Myogenin positivity discriminates ARMS from ERMS with high significance, with most ARMS containing more than 50% positive cells, while most ERMS have less than 50 percent positive cells. These numbers are in good agreement with available literature, where Myogenin has been proposed as marker to discriminate ARMS from ERMS tumors (Dias P et al., Am J Pathol. 2000). In contrast, most PDX are composed of nearly 100 percent Desmin positive cells.

Based on the limitations mentioned above, we changed the conclusion of the histological analysis on page 7-8 to:

“Impressively, both PDX and CDX show characteristic RMS architecture and a degree of MYOGENIN and DESMIN positivity, which is in line with published data showing that number of MYOGENIN positive cells discriminates ARMS from ERMS (Suppl. Figure S6A and S6B).”

6. Figure 5 is an important result. Certainly, the ability to perform direct drug testing on primary patient biopsy material could be clinically useful. There are a number of differences in the response; in general the primary LN-derived cells were more susceptible than the original PDX. Can the authors speculate on why, and what this means for their method? Were the conditions that were optimized for the PDX used for the primary tumor culture?

->Differences in response between PDX and patient-derived cultures could be generated by different aspects. First, the PDX-derived cells originated from the first relapse, whereas the direct culture was established from the second relapse (lymph node metastasis). Furthermore, these specific cells were quite sensitive towards passaging and tended to differentiate very easily which was probably an important aspect influencing the drug response.

The conditions that were used for the primary tumor-derived culture and the PDX-derived culture were slightly different, with the NB medium mixture used for the patient cells and the Adv.DMEM/F12 mixture for the PDX cells.

7. Figure 6B is confusing. What is the meaning of the triangles at right, are these opposing

gradients in concentration of the two inhibitors? Please indicated the three groups that are supposedly shown here.

-> The triangles on the right were supposed to illustrate the sensitivity towards the MEK inhibitor (red) and the AKT inhibitor (blue). We realized that the text "MEK inhibition" and "AKT inhibition" in the triangles is misleading as well as was the orientation of the triangles. We therefore changed the text to "MEKi sensitivity" and "AKTi sensitivity" and updated the Figure accordingly.

8. Figure 6, how were the # of floating cells quantified?

-> These pictures should illustrate qualitatively the morphological change upon treatment with Afuresertib. Hence, originally we did not further quantify the number of floating cells. We now repeated the experiment and included cell counts in the Figure, which is now in the supplemental part as suppl. Figure 11B.

Reviewer #3 (Expertise: Drug screening, cancer, Remarks to the Author):

This manuscript, by Manzella, Wachtel, and co-workers, describes the development of a screening platform derived from patient samples of rhabdomyosarcoma (RMS) tumors. While not a novel concept, as such screens are reported previously, this work is unusual in its complexity and completeness, as the authors went out of their way to investigate aspects of the platform in 2D, 3D, and PDX settings. Further, while the overall concept may not be new, this paper does extend the implementation of a patient-derived screening platform to a new tumor type, and the sensitivity to Akt inhibition may be translatable back into the clinic.

This paper much digs more deeply into optimized conditions for the culture of primary cancer cells compared to referenced studies. This is an extremely thorough study that may be enabling to future efforts in many tumor types. Characterization of the stromal tissue and identification of toxicity with several common conditions may serve as a warning to future researchers in related work. Better fidelity to the original sample (confirmed in this work using both histological and sequencing methods) is likely to lend more confidence in the reported results and, per the findings of the authors, to produce clones that better represent the initial tumor. An impactful finding of this study, for example, is that direct culture from patient samples is similar to use of an intermediary PDX – the requirement for *in vivo* implantation would introduce more time delays, which might limit or delay clinical benefit to identify precision medicines on a patient-by-patient basis.

We would like to thank the reviewer for his supportive comments.

In the abstract, the authors state that the responses to the screen were "surprisingly heterogeneous". Given the stated diversity in RMS subtypes (esp ERMS), it is not clear why this would be surprising.

-> We removed the term "surprisingly".

From the screen, the principal sensitivity followed-up on is that to Akt inhibitors, which was found in a select subgroup of samples. Notably, sensitivity to the AKT/PI3K axis was found in multiple similar studies referenced by the author (from references 7-11, at least 7, 8, and 10 mention sensitivity to mTor/PI3K/Akt). Is this perhaps a general trend in the technique? Or an indication of widespread sensitivity to these inhibitors? Or just a coincidence? The follow-up on the Akt inhibition is generally well done, including exploration of combinations with MEK inhibition (also identified in the screen), and is complete with both efficacy and biomarker results. This would have been further enhanced by an *in vivo* combination with the MEKi.

-> We agree that an *in vivo* study of the combination of AKT and MEK inhibitors complements our dataset. Therefore, we now performed such an additional experiment using a combination of afuresertib and trametinib. As suggested by reviewer #1, we used the two drugs in concentrations

applicable to human patients, hence afuresertib was used in a 5x lower concentration than in the *in vivo* experiment shown in Fig 6F. This novel data is depicted in Fig.6G and shows that the two drugs indeed display synergy.

As suggested by the editor, we also tested this drug combination *in vitro* with 8 different PPCs and detected in most cases a synergistic behavior of the two drugs (as determined by Bliss synergy analysis). Especially in case of some *RAS* mutant eRMS, very high synergy scores (Bliss scores 10-20) were detected. However, among the *RAS* mutant eRMS (4 cases) there is still a significant heterogeneity of responses with IC50 values of Trametinib of 4, 7.5, 32, and 238 nM. This heterogeneity is at least partially explained by the ratio of cells responding with cytotoxicity vs cytostatics (differentiation), which decreases with increasing IC50. To evaluate this further, we determined the number of Myosin Heavy Chain positive cells in the most resistant case by high throughput microscopy. This data is depicted in the new suppl. Figures 10B and C. All together, these findings again highlight the relevance of our profiling strategy for drug selection.

In addition, this reviewer noted that at least one of the mTOR compounds (Torin 2) shows a roughly similar profile to the AKT inhibitors. It would be interesting to compare that compound with the Akt-targeted compounds for at least an *in vitro* follow-up.

-> Thank you very much for this observation. However, differences in response towards torin-2 were much smaller than in case of AKT inhibitors (Remaining cell viabilities were for torin-2: 10/21 samples <30% and 20/21 samples <60%; and for afuresertib: 5/21 samples <30% and 10/21 samples <60%). Nevertheless, we still considered the suggested experiment to be of interest. Hence, we performed a detailed dose-response study of torin-2 with 7 different PPC cultures *in vitro*. The data is shown in the new suppl. Fig. 9D and demonstrates that, in contrast to MEK and AKT inhibitors, response to torin-2 is much more uniform for these 7 samples, with IC50 values all lying in a concentration range between 13 and 70 nM.

Given that a range of compounds with various inhibition patterns on various targets were used, it is curious to me that the authors chose 500 nM of all inhibitors on the original screen. This may be well above the IC50/IC90 for some compounds on their primary biochemical target, but maybe only at or near (or even below) the IC50/IC90 for others, especially given variances in cellular permeability often observed in molecules. Perhaps a better screening strategy may be to select a concentration for each inhibitor that correlates with either an achievable sustained dose *in vivo* or at least correlates to the potency of the molecule in a biochemical or cellular setting. Doing so may or may not yield different results, but might well wind up yielding even more relevant results.

->Our screening procedure was originally designed for identification of interesting sensitivities in individual cases compared to the entire set of samples. Since our platform includes establishment of culture conditions that allow continuous propagation of the cells, the idea was to select interesting drugs and perform a full dose response curve in a second step (as depicted in Supplemental Figure S9A and S9D (dose response curves of 6 drugs and Torin-2), Figure 4B/Supplemental Figure S9B (IC50 values and dose response curves for 12 drugs determined with 2D and 3D cultures) and Figures 6C and 6D (dose response curves of one MEK and two AKT inhibitors).

Overall, this is a quite good manuscript, and is recommended for publication.

Typographical/grammatical feedback:

1. Abstract, first line. Would read better if phrased as "Cancer therapy is currently shifting from..."
2. Introduction, first line (79/80) would read better as "allows cures for many previously..."
3. Discussion, line 387, Perhaps "While" would be a better word choice than "Albeit"?

-> Thank you very much for these suggestions. We changed all these sentences accordingly.

REVISION 2:

REVIEWER

COMMENTS:

Reviewer#1:

The authors have responded to my previous concerns, but unfortunately have not really addressed the critical issues raised.

In this revision, the authors have included additional information particularly concerning the in vitro to in vivo activity of combination treatments. While the data are of interest, the activity of afuresertib +trametinib is still progressive disease (using clinical criteria). Several groups have shown the activity of combined MAPK inhibitors having activity against RAS-mutant embryonal rhabdomyosarcomas (some lines are indeed very responsive), so this aspect is not particularly novel.

-> Albeit we do not see tumor regression during treatment with the combination of afuresertib and trametinib, tumor size is at least stable during the treatment period. According to the RECIST criteria, this is classified as stable disease.

We agree that activity of MAPK inhibitors and dual inhibition with PI3K inhibitors against established RMS cell lines has been described previously both by the Shipley and Fulda laboratories. However, these experiments were designed to explore the reliability of our in vitro drug profiling platform by in vivo PDX models. Hence, the fact that previously published results can be nicely confirmed presents a strong argument in favor of the accuracy of our drug profiling platform. Furthermore, our results identified heterogeneity in response towards MEK inhibitors even among the different RAS-mutant ERMS, a fact which has not been described before and which argues again for phenotypic profiling of each individual patient.

However, I feel that although a great deal of valuable information is presented (e.g. growth conditions for propagating cell lines), this work does not represent a conceptual leap forward. The authors present 21 RMS cell lines. Why 21? Are 11 fusion-negative lines adequate to represent the heterogeneity/diversity and similarly 10 fusion-positive lines - same question.

-> We used all cultures that were available in our laboratory at the time of the study. Nevertheless, we agree with the reviewer that heterogeneity of RMS (or any tumor in general) is likely never fully represented with a certain number of models, while the degree of heterogeneity/diversity for RMS specifically is currently largely unknown, apart from the distinction of fusion-negative versus fusion-positive tumors, and does not allow mathematical modeling of numbers. It is therefore exactly this enormous variability among tumors that argues for the usefulness of phenotypic profiling, since this approach assumes that, at the end, each patient's tumor represents its own entity and drug response can hardly be predicted based on surrogate markers but needs to be measured directly.

The data presented in Fig.4A is also troubling. HSP90 and proteasome inhibitors have demonstrated no activity against rhabdomyosarcoma models (or clinically), whereas bortezomid is quite active against ALL PDX models. Thus, it is unclear how the screen is of value - for example it does not separate bortezomid from the other proteasome inhibitors. Similarly, Torin 2 would be an agent

selected from the screen yet TOR kinase inhibitors appear to have less *in vivo* activity than do rapalogs (everolimus is inactive in the screen presented).

-> Translation of pre-clinical response data to efficacy in patients is indeed a difficult task that will have to be explored further in the future, likely within clinical trials. While we do not entirely exclude that also drugs with broad effects on all RMS models will offer some clinical potential, we believe that exceptional/specific *in vitro* sensitivities of individual cases represent the more interesting finding. Based on this assumption, the AKT inhibitors were more interesting for us than the drugs with general toxicity like Torin-2 or bortezomib. However, we agree that the initial screen shown in Fig.4A does not allow to distinguish effects of different proteasome and HSP90 inhibitors in great detail. For this, follow-up dose response curves are necessary, which we did for bortezomib in the co-clinical case (IC₅₀ about 10 nM; see Fig.5E) and for Carfilzomib and HSP90 in the comparative study of 2D vs 3D conditions with two cultures (Fig.4B and Suppl. Fig.S9). An IC₅₀ of 10nM for Bortezomib is in the range of IC₅₀ values previously reported for RMS cell lines (Bersani et al., Euro.J.Cancer 2008). Finally, additional parameters like data on *in vivo* availability and previous phase I toxicities will have to be considered when translating *in vitro* drug profiling data to clinical activities. We believe that the field is at its beginning and only further improvements will provide answers in the future.

For the proposed *in vitro* screen to be valuable, the authors would need to demonstrate that based on the screen they could identify multiple agents that induced tumor regressions when tested against the respective PDX.

-> We agree that more data would always be beneficial. However, due to limited resources we focused our initial *in vivo* validation on the effects of AKTi and MEKi.

Rhabdomyosarcoma is treated with essentially 6 DNA damaging agents + vincristine (V): cyclophosphamide/actinomycin D (VAC) irinotecan (I), doxorubicin, etoposide and ifosfamide + radiation. Both VAC and VI give similar response rates (65-72% objective responses), yet these agents, where tested, have quite limited activity in the *in vitro* screen. Perhaps testing of the known actives (activated cyclophosphamide or SN-38) would give more confidence in the predictions of this assay?

-> We agree that response rates among our cultures towards standard chemotherapeutic agents are somewhat lower than the 65-72% found in patients. However, our cohort contains mostly relapse samples (see suppl. Figure S1A, previous treatment) which developed resistances towards first-line treatments in the patient. Importantly, our cohort also contains pairs of diagnostic and relapse tumors from the same patient, which allow tracking of the development of drug resistance. For example, primary cultures from the tumor SJRHB13758_X1 (**diagnostic tumor**) are much more sensitive towards standard chemotherapeutics than cells from the tumor SJRHB13758_X1 (**relapse tumor from the same patient**) (see Fig.4A). The same is the case for the pair SJRHB011_X (**diagnostic**) and SJRHB011_Y (**relapse**). Taking this into account, we believe that testing of additional chemotherapeutics would not change this conclusion. Furthermore, drug concentration would likely have to be adjusted for each chemotherapeutic agent.

Nevertheless, when selecting drugs to be included in our library, we initially focused on representation of a wide range of different pathways. We agree that for future clinical translation, the focus will have to be shifted toward approved drugs and this is something that is currently ongoing, including rigorous quality checks of the compounds before application.

Reviewer#2:

The revised manuscript "High throughput drug profiling with a living biobank of primary rhabdomyosarcoma cells unravels disease heterogeneity and detects an AKT inhibitor sensitive subgroup" by Manzella et al is a substantial improvement on the original version. The new data in Figure 6 are particularly valuable. At this point, the authors have satisfied my concerns.

There are a couple of small typographical errors in Figure 5. Figure 5A: suggest "pulmonary metastases" rather than "pumonal". Also, drug "profile" rather than "profil".

-> We thank the reviewer for this positive feedback and corrected all the typos mentioned.

Reviewer#3:

Manzella, Wachtel, and co-workers present a revised manuscript that includes additional data, including, importantly, new in vivo results. I feel that the authors have adequately addressed the concerns of the reviewers and recommend that this manuscript be published.

Additional detail on two significant topics from the initial review:

- Two of the reviewers stated concerns regarding the "one-size-fits-all" 500 nM doses of the compounds in the screen, without specific regard to compound differences with respect to other pharmacology, protein binding, permeability, etc. [It is noted that this is a difficult task and consideration in many screening exercises.] While an improvement in the process can likely be realized by future consideration of such factors and perhaps better tailoring doses of each specific compound, even in the original screen, the concerns of the reviewers are likely adequately addressed.

- The new data presented in Figure 6 with the more relevant and lower doses of the combination are powerful and certainly strengthen the paper.

Suggestions for improvement/corrections:

Line 375: "proofed" likely should be "proved"

Line 439: "affect" should be "effect"

Figure 6G: I suggest to change "low dose" to the actual doses used of the compounds

-> We would like to thank reviewer #3 for the positive news and corrected the errors accordingly.